# Reinforcement Learning-Based Adaptive Position Control Scheme for Uncertain Robotic Manipulators with Constrained Angular Position and Angular Velocity

**Zhihang Xie * and Qiquan Lin**

School of Mechanical Engineering, Xiangtan University, Xiangtan 411105, China
* Correspondence: 202131540202@smail.xtu.edu.cn

**Abstract:** Aiming at robotic manipulators subject to system uncertainty and external disturbance, this paper presents a novel adaptive control scheme that uses the time delay estimation (TED) technique and reinforcement learning (RL) technique to achieve a good tracking performance for each joint of a manipulator. Compared to conventional controllers, the proposed control scheme can not only handle the system parametric uncertainty and external disturbance but also guarantee both the angular positions and angular velocities of each joint without exceeding their preset constraints. Moreover, it has been proved by using Lyapunov theory that the tracking errors are uniformly ultimately bounded (UUB) with a small bound related to the parameters of the controller. Additionally, an innovative RL-based auxiliary term in the proposed controller further minimizes the steady state tracking errors, and thereby the tracking accuracy is not compromised by the lack of asymptotic convergence of tracking errors. Finally, the simulation results validate the effectiveness of the proposed control scheme.

**Keywords:** reinforcement learning; robotic manipulator; adaptive control

## 1. Introduction

Robotic manipulators have been extensively applied in assisting or even replacing humans to perform the various tasks such as assembling [1], machine operation [2], deburring [3], drilling [4], transportation [5] and manufacturing [6,7]. In order to successfully perform such industrial tasks, it is essential to well control the motion of robotic manipulators. However, the highly nonlinear and uncertain dynamics of robotic manipulators result in the inapplicability of many traditional controllers designed for linear systems such as linear quadratic control (LQC) [8,9] and linear H∞ control [10,11]. Hence, many researchers have been motivated to place efforts on designing advanced controllers for robotic manipulators.

State feedback linearization requiring a known dynamic model can transform the nonlinear system to a linear form [12]. However, the dynamic model of robotic manipulators is always uncertain because of unknown external disturbances and uncertain system parameters. To handle the issue of uncertainty, many efforts have been made such as sliding mode control (SMC) [13–18], fuzzy logic system (FLS)-based control [19–21], neural network (NN)-based control [22–26], and disturbance observer (DOB)-based control [27–29]. More precisely, Zhang et al. [13] propose a fixed time sliding mode control for uncertain robotic manipulators in which a conservative switching gain requiring the upper bound of lumped uncertainty is used. In [21], an adaptive controller-based T-S (Takagi-Sugeno) fuzzy system is designed, and the modified T-S fuzzy system can efficiently approximate the unknown model of robotic manipulators. Hu et al. [26] present a multiple-layer neural network-based controller that can achieve a high accuracy of motion control of robotic manipulators subject to unknown disturbances. In [28], a high-order-sliding-mode differen-

tiator (HOSMD)-based estimator is designed to compensate the mismatched uncertainties without the need of assuming a bounded uncertainty.

Besides the uncertainty of the dynamic model, state constraint is also a common problem of robotic manipulators, which has attracted much attention. For example, Sun et al. [30] proposed an adaptive neural network control scheme considering full-state constraints of robotic manipulators. In [31], an adaptive fuzzy control scheme that can guarantee the constrained output is presented. Yu et al. [32] designed an adaptive fuzzy control scheme working with a disturbance observer for the manipulators with full-state constraints. Nevertheless, those controllers can only achieve either the constrained tracking error of joint angle or the constrained error between the actual angular velocity and the virtual control signal. In other words, controllers in [30–32] do not guarantee the angular position and angular velocity for each joint of the robotic manipulator, within the preset constraints. Recently, Yang et al. [33] designed a new adaptive control scheme that can guarantee the angular position for each joint of a robotic manipulator to never exceed the preset constraints. However, the constrained angular velocity of each joint is not guaranteed.

In the light of reviewing the existing literature, the following issues need to be further solved:

- In order to safely perform robotic manipulators, both the angular position and angular velocity of each joint of robotic manipulators should be controlled to not exceed the preset constraints. More precisely, the angular position (rotation angle) of each joint should be always within a reasonable range to have no risk on physically breaking the joint. Similarly, the angular velocity of each joint should not exceed its maximum related to the maximum rotational speed of the driving motor;
- For some existing controllers (e.g., [9,23,26,30–32]), the tracking accuracy could be compromised due to the bounded result of tracking errors and uncertainty-estimation errors. Therefore, it is needed to avoid the loss of tracking accuracy caused by the lack of asymptotic convergence of tracking errors.

To handle the above issues, this paper proposes a new adaptive control scheme that utilizes time delay estimation (TDE) and reinforcement learning (RL) for *n*-link robotic manipulators. In the proposed control scheme, the multiple-input-and-multiple-output (MIMO) robotic system is initially decomposed into *n* single-input-single-output (SISO) subsystems by TDE. Each subsystem is with an unknown bounded TDE error. After that, the novel virtual control law for each subsystem is designed, which can not only achieve the boundness of tracking errors for each joint in the presence of TDE error, but can also guarantee both the angular position and angular velocity for each joint to be not exceeding the preset constraints. To improve the tracking accuracy, an RL-based term in the virtual control law is designed, which automatically learns the optimal parameters of the controller in the different system states. RL is an artificial intelligence technique that gradually explores the optimal policy by interacting with the environment, which has attracted many interests in the control of robotic manipulators such as [34–36]. Particularly, the RL-based term in this paper is designed to avoid the violation of the boundness of tracking errors, even if a bad policy is tried by RL, which ensures a safe environment for implementing RL. The tracking errors are proven to be uniformly ultimately bounded (UUB) via Lyapunov theory. Simulation results indicate the proposed control scheme can achieve a high tracking accuracy in the presence of model uncertainty and unknown disturbances.

The major merits of this paper include the following points:

- Compare with some existing research [30–33], in addition to the basic achievement of the uniformly ultimately bounded (UUB) tracking error for each joint in the presence of TDE error; the control scheme can guarantee both the angular position and angular velocity for each joint to be not exceeding the preset constraints;
- The novel adaptive gain in (13) results in the smooth control torques to reduce the chattering effect caused by switching term in (10). Meanwhile, an RL-based term

can effectively improve the tracking accuracy, which thereby reduces the possible steady-state tracking errors caused by the lack of asymptotic convergence;

- The mathematical expression of the controller is simple, meanwhile, any prior knowledge of upper bounds caused by an imprecise model are unnecessary in our control scheme.

The rest of this paper is organized as follows: in Section 2, the dynamics model of n-link robotic manipulators is given, and the control objective is described. In Section 3, the RL-based adaptive control scheme is proposed and the proof of stability is given. In Section 4, the numerical simulation is conducted to verify the effectiveness of the proposed controller. The conclusion is given in Section 5.

## 2. Dynamical Model and Problem Statement

The dynamic model of n-link robot manipulators is shown as the following:

$$M(q(t))\ddot{q}(t) + C(q(t), \dot{q}(t))\dot{q}(t) + G(q(t)) + F(\dot{q}(t)) = \tau(t) + \tau_d(t) \tag{1}$$

where $M(q(t)) \in \mathrm{R}^{n \times n}$ is the inertia matrix, $q(t) = [q_1(t), q_2(t), .., q_i(t), .., q_n(t)]^T \in \mathrm{R}^n$ is the vector of angular positions of joints of manipulator. $C(q(t), \dot{q}(t)) \in \mathrm{R}^{n \times n}$ is the Coriolis and centrifugal matrix. $G(q(t)) \in \mathrm{R}^n$ is the gravity vector. $F(\dot{q}(t)) \in \mathrm{R}^n$ is the vector of friction. $\tau(t) \in \mathrm{R}^n$ is the vector of torques applied on the joints. $\tau_d(t) \in R^n$ is the external disturbance.

The model (1) can be further written as (2) to indicate the system uncertainty.

$$\ddot{q}(t) = M(q(t))^{-1}\left[-C(q(t), \dot{q}(t))\dot{q}(t) - G(q(t)) - F(\dot{q}(t)) + \tau_d(t)\right] + \left[M(q(t))^{-1} - \hat{M}(q(t))^{-1}\right]\tau(t)$$
$$+\hat{M}(q(t))^{-1}\tau(t) = \Gamma(t) + \hat{M}(q(t))^{-1}\tau(t) \tag{2}$$

where $\hat{M}(q(t))$ is the estimation of $M(q(t))$. $\Gamma(t) = M(q(t))^{-1}\left[-C(q(t), \dot{q}(t))\dot{q}(t) - G(q(t))\right.$ $\left.-F(\dot{q}(t)) + \tau_d(t)\right] + \left[M(q(t))^{-1} - \hat{M}(q(t))^{-1}\right]\tau(t)$ is the system uncertainty.

The vector of error between the desired angular position and actual angular position is defined in (3):

$$e(t) = q(t) - q_d \tag{3}$$

where $q_d = [q_{d1}, q_{d2}, \ldots, q_{dn}]^T \in R^n$ is the vector of desired angular position of joints. $e = [e_1, e_2, \ldots, e_n]^T \in R^n$ is the tracking error vector.

The main control objective is to design the torque $\tau(t)$ that can drive all the joints of the robotic manipulator system (1) to approach their desired angular positions. Meanwhile, both the angular positions and angular velocities for each joint should be guaranteed to not exceed the given constraints. The control objective can be described by (4)–(6).

$$||e(t)|| \leq \sigma, \ \forall \, t \geq 0 \tag{4}$$

$$|q_i(t)| < \varepsilon_i, \ \forall \, t \geq 0, \ i = 1, 2, .., n \tag{5}$$

$$\left|\dot{q}_i(t)\right| < \Lambda_i, \ \forall \, t \geq 0, \ i = 1, 2, .., n \tag{6}$$

where $\sigma^* > \sigma \geq 0$, and $\sigma^* > 0$ is a positive constant, $\varepsilon_i > 0$ is a positive constant referring to the angular position constraint of the $i^{th}$ joint, and $\Lambda_i > 0$ is a positive constant, meaning the angular velocity constraint of the $i^{th}$ joint.

**Remark 1.** *The positive constant $\sigma^*$ is related to the initial state of the system and parameters of the controller. $\sigma$ reflects the tracking accuracy. Notably, $\sigma(t \to \infty) = 0$ means the asymptotic convergence of tracking errors.*

**Remark 2.** *In this paper, we consider the angular position tracking problem for each joint of the manipulator. Hence, $q_{di}$ is a constant for $i = 1, 2, .., n.$, which means $\dot{q}_{di} = 0$. The angular trajectory tracking problem will be considered in our future work.*

### 3. Controller Design and Stability Analysis

In this part, the adaptive RL-based controller working with TDE is developed. After that, the stability is proven by using Lyapunov theory.

*3.1. Controller Design*

The TDE technique is applied to handle the system uncertainty in (2):

$$\hat{\boldsymbol{\Gamma}}(t) \approx \boldsymbol{\Gamma}(t - L) = \ddot{\boldsymbol{q}}(t - L) - \hat{\boldsymbol{M}}^{-1}(\boldsymbol{q}(t - L))\boldsymbol{\tau}(t - L) \tag{7}$$

where $\hat{\boldsymbol{\Gamma}}(t) = [\hat{\Gamma}_1, \hat{\Gamma}_2, .., \hat{\Gamma}_n]^T \in R^n$ is the estimate of $\Gamma(t)$. $L > 0$ is the sampling time of TDE.

**Lemma 1.** *[14] The TDE error of robotic manipulator (2) is bounded such that $|\Gamma_i(t) - \hat{\Gamma}_i(t)| \leq \Gamma_i^*$ (for $i = 1, 2, \ldots, n$) if the following condition is satisfied:*

$$\left\| \boldsymbol{I} - \boldsymbol{M}^{-1}(\boldsymbol{q}(t))\hat{\boldsymbol{M}}(\boldsymbol{q}(t - L)) \right\|_2 < 1 \tag{8}$$

where $\Gamma_i^*$ is an unknown positive constant.

The control law working with TDE technique is designed as follows:

$$\boldsymbol{\tau}(t) = \hat{\boldsymbol{M}}(\boldsymbol{q}(t))\left[ -\hat{\boldsymbol{\Gamma}}(t) + \boldsymbol{u}(t) \right] \tag{9}$$

where $\boldsymbol{u}(t) = [u_1(t), u_2(t), \ldots, u_n(t)]^T \in R^n$ is the virtual control law.

The virtual control law in (9) is designed as follows:

$$u_i = -\hat{d}_i sgn(\dot{q}_i) - \frac{1}{1 + k_{yi}C_i} \left[ k_{pi}e_i + k_{di}\dot{q}_i + k_{si}A_iq_i + k_{si}B_iz_i + \Lambda_i\dot{q}_i + \Lambda_i\Lambda_i tanh\left(\frac{e_i}{\mathbb{C}_i}\right) \right] \tag{10}$$
$$ti = 1, 2, \ldots, n$$

where $k_{pi}$, $k_{si}$, $k_{di}$, $\mathbb{C}_i$ and $k_{yi}$ are the positive constants determined by users. $\Lambda_i > 0$ is a positive variable determined by the fuzzy reinforcement learning mechanism.

$$A_i = \frac{\varepsilon_i^2 z_i^2}{\left(\varepsilon_i^2 - q_i^2\right)^2}, \ B_i = \frac{\varepsilon_i^2}{\varepsilon_i^2 - q_i^2}, \ C_i = \frac{2\Lambda_i^2}{\left(\Lambda_i^2 - \dot{q}_i^2\right)^2} \tag{11}$$

where $\varepsilon_i > 0$ is the restricted upper bound of angular position of the $i^{th}$ joint. $\Lambda_i > 0$ is the restricted upper bound of angular velocity of the $i^{th}$ joint.

And the variable $z_i$ is defined as follows:

$$z_i = e_i + \int_0^t \eta_i(\Theta)d\Theta, \ \eta_i = -\beta_iz_i \tag{12}$$

where $\beta_i$ is a positive constant.

The $\hat{d}_i$ is used to handle the bounded TDE error and the update law of $\hat{d}_i$ is designed as follows.

$$\dot{\hat{d}}_i = \begin{cases} \psi_i(1 + k_{yi}C_i)|\dot{q}_i|, \ if\left(\hat{d}_i \leq 0\right) or \left(\Omega_i > \overline{\Omega}_i\right) \\ \\ -\psi_i\frac{\delta_i}{(1+k_{yi}C_i)|\dot{q}_i|}, \ if\left(\hat{d}_i > 0\right) and \left(\Omega_i \leq \overline{\Omega}_i\right) \end{cases} \tag{13}$$

where $\psi_i$, $\delta_i$ and $\overline{\Omega}_i$ are positive constants. The variable $\Omega_i$ is defined as (14).

$$\Omega_i = \frac{1}{2}\dot{q}_i^2 + \frac{1}{2}k_{pi}e_i^2 + \frac{1}{2}k_{si}\frac{\varepsilon_i^2 z_i^2}{\varepsilon_i^2 - q_i^2} + k_{yi}\frac{\dot{q}_i^2}{\Lambda_i^2 - \dot{q}_i^2} + \Lambda_i \mathbb{C}_i \ln\left[\cosh\left(\frac{e_i}{\mathbb{C}_i}\right)\right] \tag{14}$$

**Remark 3.** *$\hat{d}_i$ is to guarantee the stability in the presence of the bounded TDE errors. A conservative update law of $\hat{d}_i$ is to monotonously increase the value of $\hat{d}_i$ such that $\dot{\hat{d}}_i = \psi_i(1 + k_{yi}C_i)|\dot{q}_i|$. Although such adaptive law can achieve the bounded tracking errors and angular velocities, the great value of $\hat{d}_i$ could result in the chattering effect on the calculated control torques. Therefore, a novel adaptive law for $\hat{d}_i$ shown in (13) is proposed to mitigate the chattering effect by decreasing the value of $\hat{d}_i$ without the breach of stability of system. The proof will be given later. Moreover, $\hat{d}_i > 0$ holds because $\dot{\hat{d}}_i \leq 0$ leads to $\dot{\hat{d}}_i = \psi_i(1 + k_{yi}C_i)|\dot{q}_i| \geq 0$.*

**Remark 4.** *Similar to [33], the terms $A_i q_i$ and $B_i z_i$ in (10) are to guarantee the angular position of each joint of manipulator to not exceed the preset constraint $\pm \varepsilon_i$. While the term $\frac{1}{1+k_{yi}C_i}$ in (10) is to guarantee the angular velocity of each joint to not exceed the preset constraint $\pm \Lambda_i$, which was not achieved in [33]. The proof will be detailed later.*

*3.2. Stability Analysis*

**Theorem 1.** *If the initial angular position and velocity of all joints are within their preset constraints such that $|q_i(0)| < \varepsilon_i$ and $|\dot{q}_i(0)| < \Lambda_i$, and (8) in lemma 1 holds, the control law consisting of (7) and (9)–(14) can achieve the uniformly ultimately bounded tracking errors of robotic manipulator system (1). Meanwhile, the angular velocity and angular position of each joint of manipulator are within the preset constraints such that $|q_i(t)| < \varepsilon_i$ and $|\dot{q}_i(t)| < \Lambda_i$, $\forall\, t > 0$. Namely, the control target (4)–(6) is achieved.*

**Proof.** By inserting (9) into (2) and using the fact of $\hat{M}^{-1}\hat{M} = I$ with $I = diag([1,1,..,1]) \in R^{n \times n}$, the MIMO robotic manipulator system can be decoupled into $n$ uncertain SISO subsystems (15).

$$\ddot{q}_i(t) = u_i(t) + d_i(t)\quad i = 1, 2, \ldots, n \tag{15}$$

where $q_i \in q$ is the angular position of the $i^{th}$ joint. $u_i$ is designed in (10). $d_i(t) = \Gamma_i(t) - \hat{\Gamma}_i(t)$ is the TDE error.

The following Lyapunov candidate is designed:

$$V = \sum_{i=1}^{n} V_i = \sum_{i=1}^{n}\left[\frac{1}{2}\dot{q}_i^2 + \frac{1}{2}k_{pi}e_i^2 + \frac{1}{2\psi_i}\tilde{d}_i^2 + k_{si}\frac{\varepsilon_i^2 z_i^2}{2(\varepsilon_i^2 - q_i^2)} + k_{yi}\frac{\dot{q}_i^2}{\Lambda_i^2 - \dot{q}_i^2} + \Lambda_i\,\Lambda_i \mathbb{C}_i \ln\left(\cosh\left(\frac{e_i}{\mathbb{C}_i}\right)\right)\right] \tag{16}$$

where $\tilde{d}_i = \Gamma_i^* - \hat{d}_i$. $\Gamma_i^*$ is the upper bound TDE error defined in Lemma 1.

**Remark 5.** *Clearly, $k_{si}\frac{\varepsilon_i^2 z_i^2}{2(\varepsilon_i^2 - q_i^2)} > 0$ holds as long as $|q_i| < \varepsilon_i$. $k_{yi}\frac{\dot{q}_i^2}{\Lambda_i^2 - \dot{q}_i^2} > 0$ as long as $|\dot{q}_i| < \Lambda_i$. Furthermore, $\Lambda_i \mathbb{C}_i ln\left(cosh\left(\frac{e_i}{\mathbb{C}_i}\right)\right) \geq 0$ holds because of the fact of $\cosh(\cdot) \geq 1$ and the fact of $ln(x) \geq 0$ with $x \geq 1$. Therefore, the Lyapunov candidate (16) is positive defined as long as $|q_i| < \varepsilon_i$ and $|\dot{q}_i| < \Lambda_i$.*

Taking the derivative of the Lyapunov function (16) with respect to the time $t$ and using (11), (12) and (15), we have:

$$\dot{V} = \sum_{i=1}^{n} [\dot{q}_i \ddot{q}_i + k_{pi} e_i \dot{q}_i - \frac{1}{\psi_i} \widetilde{d}_i \dot{\hat{d}}_i + k_{si} A_i q_i \dot{q}_i + k_{si} B_i z_i \dot{z}_i + k_{yi} C_i \dot{q}_i \ddot{q}_i + \Lambda_i \Lambda_i tanh\left(\frac{e_i}{\mathbb{C}_i}\right) \dot{q}_i]$$

$$= \sum_{i=1}^{n} \dot{q}_i \left[ (1 + k_{yi} C_i)(u_i + d_i) + k_{pi} e_i + k_{si} A_i q_i + k_{si} B_i z_i + \Lambda_i \Lambda_i tanh\left(\frac{e_i}{\mathbb{C}_i}\right) \right] \qquad (17)$$

$$- \sum_{i=1}^{n} \left( \frac{1}{\psi_i} \widetilde{d}_i \dot{\hat{d}}_i + k_{si} B_i z_i \eta_i \right)$$

Substituting (10) and into (17) with the fact of $C_i > 0$ as long as $|\dot{q}_i| < \Lambda_i$:

$$\dot{V} = \sum_{i=1}^{n} \dot{q}_i (1 + k_{yi} C_i) \left[ -\hat{d}_i sgn(\dot{q}_i) + d_i \right] - \sum_{i=1}^{n} \frac{1}{\psi_i} \widetilde{d}_i \dot{\hat{d}}_i$$

$$- \sum_{i=1}^{n} k_{si} B_i \beta_i z_i^2 - \sum_{i=1}^{n} (\Lambda_i + k_{di}) \dot{q}_i^2$$

$$\leq \sum_{i=1}^{n} |\dot{q}_i| (1 + k_{yi} C_i) \left( \Gamma_i^* - \hat{d}_i \right) - \sum_{i=1}^{n} \frac{1}{\psi_i} \widetilde{d}_i \dot{\hat{d}}_i \qquad (18)$$

$$= \sum_{i=1}^{n} \left[ |\dot{q}_i| (1 + k_{yi} C_i) - \frac{1}{\psi_i} \dot{\hat{d}}_i \right] \left( \Gamma_i^* - \hat{d}_i \right)$$

Therefore, for each $V_i$, we can derive (19) by combining (13) and (18):

$$\dot{V}_i \leq \begin{cases} 0 \, , \, if\left( \hat{d}_i \leq 0 \right) \, or \, \left( \Omega_i \geq \overline{\Omega}_i \right) \\ \\ \mathfrak{x}_i \left( \Gamma_i^* - \hat{d}_i \right), \, if\left( \hat{d}_i > 0 \right) \, and \, \left( \Omega_i < \overline{\Omega}_i \right) \end{cases} \qquad (19)$$

where $\mathfrak{x}_i = |\dot{q}_i| (1 + k_{yi} C_i) + \frac{\delta_i}{|\dot{q}_i| (1 + k_{yi} C_i)}$. Clearly, $\mathfrak{x}_i \geq 2\sqrt{\delta_i} > 0$ holds.

In addition, the $V_i$ defined in (16) can be written as:

$$V_i = \Omega_i + \frac{1}{2\psi_i} \widetilde{d}_i^2 \qquad (20)$$

where $\Omega_i$ is defined in (14).

To prove the Lyapunov function, $V_i$ is bounded by a positive constant and we assume a sufficiently large constant $V_i^*$. Clearly, the sufficiently large $V_i^*$ requires at least one of the terms ($\Omega_i$ or $\widetilde{d}_i^2$) to be sufficiently large. If $\Omega_i$ is sufficiently large such that $\Omega_i > \overline{\Omega}_i$, then $\dot{V}_i \leq 0$ holds according to (19). If $\widetilde{d}_i^2 = \left( \Gamma_i^* - \hat{d}_i \right)^2$ is sufficiently large such that $\widetilde{d}_i^2 > (2\Gamma_i^*)^2$, then $\hat{d}_i > 3\Gamma_i^*$ will hold because of the facts of $\Gamma_i^* > 0$ (Lemma 1) and $\hat{d}_i \geq 0$ (remark 3), which further means $\Gamma_i^* - \hat{d}_i < 0$ holds. As a result, according to (19) and the fact of $\mathfrak{x}_i \geq 2\sqrt{\delta_i} > 0$, $\dot{V}_i \leq 0$ can hold by $\Omega_i > \overline{\Omega}_i$ or $\widetilde{d}_i^2 > (2\Gamma_i^*)^2$, which means $\dot{V}_i \leq 0$ holds if $V_i \geq \overline{\Omega}_i + \frac{1}{2\psi_i} (2\Gamma_i^*)^2$ holds. It is thereby easy to conclude that $V_i$ is bounded by $V_i^*$ such that $V_i \leq V_i^* = \max\left\{ V_i(0), \overline{\Omega}_i + \frac{1}{2\psi_i} (2\Gamma_i^*)^2 \right\}$.

Therefore, the tracking error of the $i^{th}$ subsystem is bounded because of the fact of $\frac{1}{2} k_{pi} e_i^2 \leq V_i$.

$$|e_i| \leq \sqrt{\frac{2}{k_{pi}} \max\{V_i(0), \, (\overline{\Omega}_i + \frac{1}{2\psi_i} (2\Gamma_i^*)^2)\}} \, i = 1, 2, .., n \qquad (21)$$

Then, the norm of the vector of the tracking error is thereby bounded.

$$||e(t)|| \leq \sqrt{\sum_{i=1}^{n} \frac{2}{k_{pi}} \max\{V_i(0), [\overline{\Omega}_i + \frac{1}{2\psi_i} (2\Gamma_i^*)^2]}} \qquad (22)$$

Moreover, the bounded $V_i$ implies all terms in $V_i$ are bounded. The (23) therefore holds, which means $|q_i| < \varepsilon_i$ and $|\dot{q}_i| < \Lambda_i$ hold.

$$\frac{\varepsilon_i^2 z_i^2}{2(\varepsilon_i^2 - q_i^2)}, \; \frac{\dot{q}_i^2}{\Lambda_i^2 - \dot{q}_i^2} \in L_\infty i = 1, 2, \ldots, n \tag{23}$$

□

### 3.3. Fuzzy Q Reinforcement Learning Mechanism Determining Parameters of Controller

In this section, a fuzzy Q reinforcement learning mechanism is designed to tune the parameter $\Lambda_i$ to improve the tracking accuracy. The motivations of RL are detailed in remark 7.

**Lemma 2.** *If (24) holds, the tracking error $e_i$ defined in (3) will asymptotically converge to zero with the converging rate satisfying $|\dot{q}_i| < \Lambda_i$.*

$$0 = \dot{q}_i + \Lambda_i tanh\left(\frac{e_i}{\mathbb{C}_i}\right), \; i = 1, 2, ..n \tag{24}$$

**Proof.** A simple Lyapunov function is given in (25).

$$\mathcal{V}_i = \frac{1}{2} e_i^2 \tag{25}$$

Combining (24) and derivative of (25), we can obtain (26).

$$\dot{\mathcal{V}}_i = -\Lambda_i e_i tanh\left(\frac{e_i}{\mathbb{C}_i}\right) \le -\Lambda_i \mathbb{C}_i tanh^2\left(\frac{e_i}{\mathbb{C}_i}\right) \le 0 \tag{26}$$

$\dot{\mathcal{V}}_i \le 0$ implies $\mathcal{V}_i \in L_\infty$. And then, (27) can be obtained by integration on both sides of (26).

$$\mathcal{V}_i(\infty) - \mathcal{V}_i(0) \le -\Lambda_i \mathbb{C}_i \int_0^\infty tanh^2\left(\frac{e_i}{\mathbb{C}_i}\right) \tag{27}$$

In the light of Barbalet's Lemma, (27) implies $tanh\left(\frac{e_i(\infty)}{\mathbb{C}_i}\right) = 0$, which means $e_i(\infty) = 0$. Moreover, it is clear that $|\dot{q}_i| = |\Lambda_i tanh\left(\frac{e_i}{\mathbb{C}_i}\right)| < \Lambda_i$ due to the fact of $tanh(\cdot) < 1$.

□

**Remark 6.** *The asymptotic convergence of tracking errors in Lemma 2 is stronger than the boundedness of tracking errors that we achieved in Theorem 1, which means a better tracking accuracy. It is because asymptotic convergence means $e_i$ is eventually going to zero, while boundedness only implies $|e_i|$ bounded by a positive constant. Therefore, the proposed controller can be improved with a better tracking accuracy by finding the optimal parameters of the proposed controller that are able to minimize $\left|\dot{q}_i + \Lambda_i tanh\left(\frac{e_i}{\mathbb{C}_i}\right)\right|$.*

The converging behaviour of tracking errors of (24) is visualized in Figure 1.

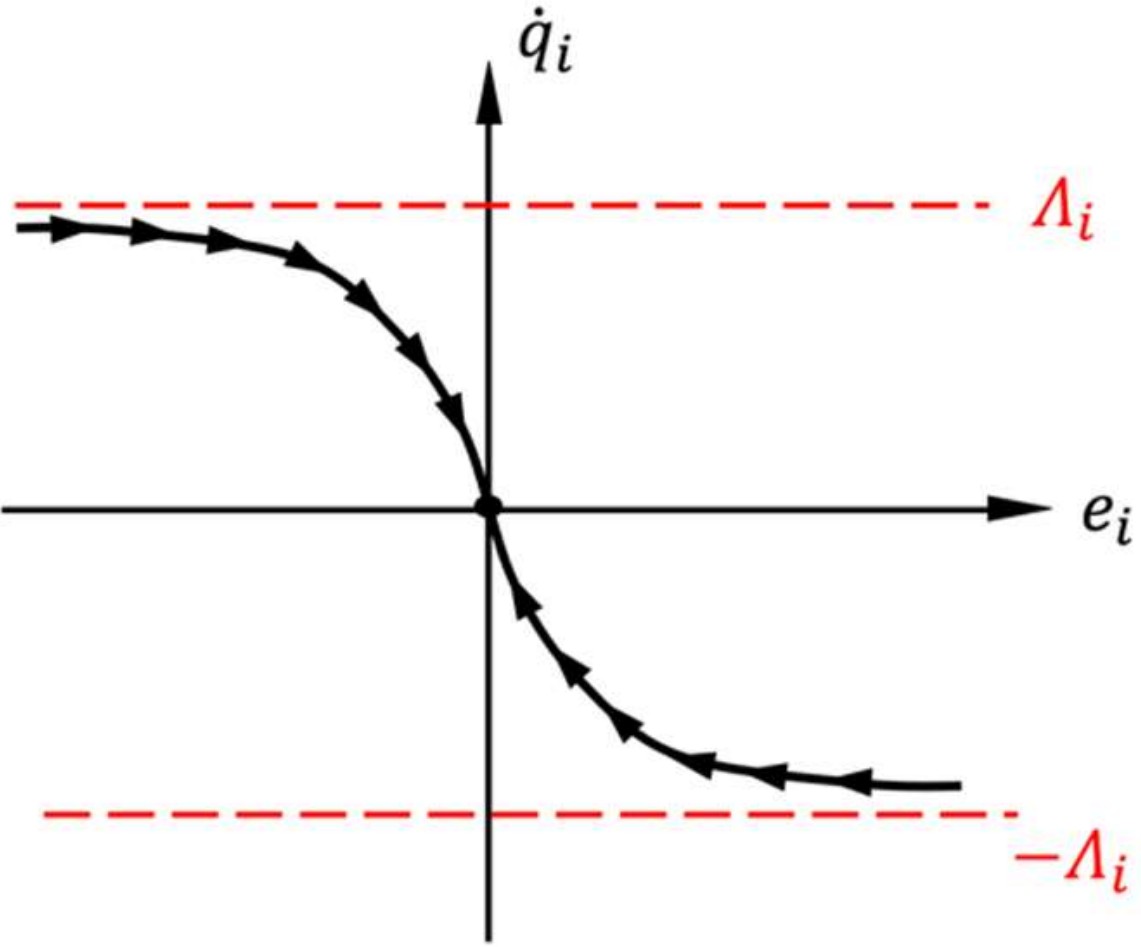

**Figure 1.** Converging behaviour of tracking error satisfying (24).The red line is the velocity constraint, and the black line with arrow is the converging trajectory.

**Lemma 3.** *For a moment $t^*$, if the system states $\left(q_i(t^*), \dot{q}_i(t^*)\right)$ satisfy (5) and (6) as well as $\dot{q}_i(t^*) + \Lambda_i tanh\left(\frac{e_i(t^*)}{\mathbb{C}_i}\right) \neq 0$ holds, there always exists the optimal parameter $\Lambda_i^* > 0$ to decrease $\left(\dot{q}_i + \Lambda_i tanh\left(\frac{e_i}{\mathbb{C}_i}\right)\right)^2$ . Namely, (28) holds.*

$$\frac{d\left[\left(\dot{q}_i + \Lambda_i tanh\left(\frac{e_i}{\mathbb{C}_i}\right)\right)^2\right]}{dt}\Big|_{t=t^*} < 0 \tag{28}$$

**Proof.** Combining (10) and (15), we can obtain the 1st derivative of $\dot{q}_i + \Lambda_i tanh\left(\frac{e_i}{\mathbb{C}_i}\right)$ with respect to time, shown as (29).

$$\frac{d\left(\dot{q}_i + \Lambda_i tanh\left(\frac{e_i}{\mathbb{C}_i}\right)\right)}{dt} = -\hat{d}_i sgn(\dot{q}_i) + d_i(t) + \frac{\Lambda_i}{\mathbb{C}_i}\frac{\partial tanh\left(\frac{e_i}{\mathbb{C}_i}\right)}{\partial \frac{e_i}{\mathbb{C}_i}}\dot{q}_i - \frac{1}{1+k_{yi}C_i}\left[k_{pi}e_i + k_{di}\dot{q}_i + k_{si}A_iq_i + k_{si}B_iz_i\right]$$
$$- \frac{1}{1+k_{yi}C_i}\left[\Lambda_i\,\dot{q}_i + \Lambda_i\Lambda_i tanh\left(\frac{e_i}{\mathbb{C}_i}\right)\right] \tag{29}$$

Then, we define a negative changing rate of $\left(\dot{q}_i + \Lambda_i tanh\left(\frac{e_i}{\mathbb{C}_i}\right)\right)^2$ at moment $t^*$ in (30).

$$\frac{d}{dt}[\dot{q}_i(t) + \Lambda_i tanh\left(\frac{e_i(t)}{\mathbb{C}_i}\right)]^2|_{t=t^*} < 0 \tag{30}$$

Using (5), (6), (29) and (30), a solution of $\Lambda_i^*$ satisfying (30) is obtained in (31).

$$\Lambda_i^* > \frac{1}{|\dot{q}_i + \Lambda_i tanh\left(\frac{e_i}{\mathbb{C}_i}\right)|}[(1 + k_{yi}C_i)(\hat{d}_i + \Gamma_i^* + \frac{\Lambda_i^2}{\mathbb{C}_i}) + k_{pi}|e_i| + k_{di}\Lambda_i + k_{si}|A_i|\varepsilon_i + k_{si}|B_i||z_i|] \tag{31}$$

Notably, $\hat{d}_i$, $|e_i|$, $|z_i|$, $|A_i|$ and $|B_i|$ are all bounded because of the bounded Lyapunov function (16). Therefore, a finite real solution of $\Lambda_i^*$ satisfying (31) exists as long as $\dot{q}_i + \Lambda_i tanh\left(\frac{e_i}{\mathbb{C}_i}\right) \neq 0$.
$\square$

**Remark 7.** *According to Lemma 3 and Remark 6, the optimal parameters $\Lambda_i^*$ leading to a decrease of $|\dot{q}_i + \Lambda_i tanh\left(\frac{e_i}{\mathbb{C}_i}\right)|$ at the moment $t^*$ can improve the tracking accuracy. Although a large enough $\Lambda_i$ can be selected to satisfy (31), an inappropriately large $\Lambda_i$ could lead to a significant chattering of control torques and thereby compromise the tracking accuracy. Moreover, an optimal $\Lambda_i$ is hard to be deterministically found due to the complexity of the system and the unknown TDE errors. Therefore, a fuzzy Q RL mechanism is designed to automatically determine the optimal $\Lambda_i^*$.*

Fuzzy Q learning is a common version of RL applicable on continuous systems, which can explore the optimal policy by interacting with the environment [37]. In this paper, the $\Lambda_i$ is to be tuned by a fuzzy Q learning mechanism according to the tracking error $e_i$ and the angular velocity $\dot{q}_i$. The linguistic rules to determine $\Lambda_i$ can be given as the following form:

$$\text{IF } e_i(\mathbb{k}) \text{ is } \mathcal{L}_{e,i} \text{ AND } \dot{q}_i(\mathbb{k}) \text{ is } \mathcal{L}_{\dot{q},i}, \text{ THEN } \Lambda_i(\mathbb{k}) \text{ is } \mathcal{L}_{\Lambda,i} \tag{32}$$

$\mathcal{L}_{e,i}$, $\mathcal{L}_{\dot{q},i}$ and $\mathcal{L}_{\Lambda,i}$ are the linguistic descriptions of tracking error $e_i$, angular velocity $\dot{q}_i$ and parameter $\Lambda_i$ respectively. $\mathbb{k}$ is the current moment. The linguistic description could be "small", "medium" and "big". As a result, an example of linguistic rule could be: IF $e_i(\mathbb{k})$ is *small* AND $\dot{q}_i(\mathbb{k})$ is *small*, THEN $\Lambda_i(\mathbb{k})$ is *small*.
Some intuitive linguistic rules can be given as:

IF $e_i(\mathbb{k})$ is small, and $\dot{q}_i(\mathbb{k})$ is small THEN $\Lambda_i(\mathbb{k})$ is small
IF $e_i(\mathbb{k})$ is large, and $\dot{q}_i(\mathbb{k})$ is large THEN $\Lambda_i(\mathbb{k})$ is large
IF $e_i(\mathbb{k})$ is large, and $\dot{q}_i(\mathbb{k})$ is small THEN $\Lambda_i(\mathbb{k})$ is large
IF $e_i(\mathbb{k})$ is small, and $\dot{q}_i(\mathbb{k})$ is large THEN $\Lambda_i(\mathbb{k})$ is small

**Remark 8.** *To carry out the linguistic inference shown by (32) in a numerical form, fuzzy logic inference is required. In detail, initially, numerical variables $e_i(\mathbb{k})$ and $\dot{q}_i(\mathbb{k})$ at the moment $\mathbb{k}$ are fuzzified to the firing rates of the linguistic descriptions by the triangular membership function shown in Figure 2. After that, a group of firing rates of fuzzy rules are obtained by fuzzy reasoning. Then, the numerical values of $\Lambda_i(\mathbb{k})$ are calculated by the defuzzification according to the firing rates of all fuzzy rules and the numerical value of the action corresponding to each fuzzy rule.*

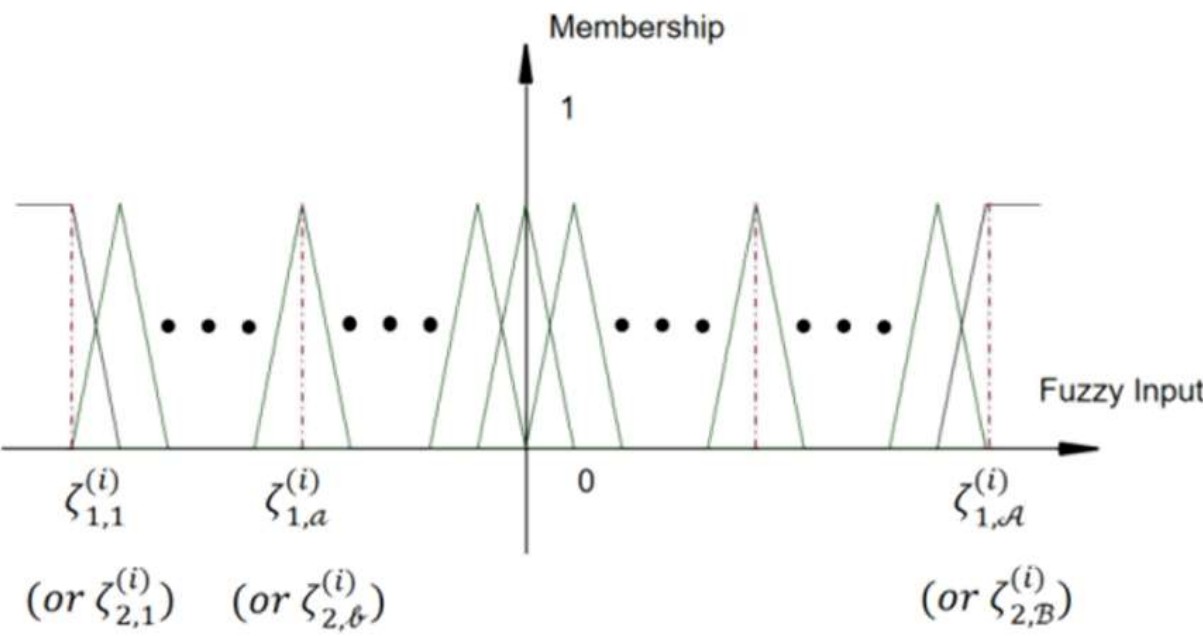

**Figure 2.** Membership function of fuzzy input of the $i^{th}$ subsystem.

The parameters of membership function shown in Figure 2 are defined as follows:

$$Lin(e_i) = \left\{ \zeta_{1,1}^{(i)}, \ldots, \zeta_{1,a}^{(i)}, \ldots, \zeta_{1,a}^{(i)} \right\}, \ a = 1, 2, \ldots, a \tag{33}$$

$$Lin(\dot{q}_i) = \left\{ \zeta_{2,1}^{(i)}, \ldots, \zeta_{2,b}^{(i)}, \ldots, \zeta_{2,b}^{(i)} \right\}, \ b = 1, 2, \ldots, b \tag{34}$$

where $a$ is the number of fuzzy sets ($\zeta_{1,a}$) for the fuzzy input $e_i$, and $b$ is the number of fuzzy sets ($\zeta_{1,b}$) for the fuzzy input $\dot{q}_i$.

The $n$-th fuzzy rule in fuzzy Q learning is defined as follows:

$$R_{n,i}: \ \text{IF } s_{1,i}^{(k)} \text{ is } L_{1,i}^{(k)} \text{ and } s_{2,i}^{(k)} \text{ is } L_{2,i}^{(n)} \text{ and } \ldots\ldots \text{ and } s_{m,i}^{(k)} \text{ is } L_{m,i}^{(n)}, \ \text{THEN } u_i^{(n)} \in \mathfrak{U}_{i,n} \text{ that } u_i^{(n)} = u_{i,1}^{(n)} \text{ with}$$
$$q_{\dot{q}i}^{(k)}(n,1) \text{ or } u_i^{(n)} = u_{i,2}^{(n)} \text{ with } q_{\dot{q}i}^{(k)}(n,2) \text{ or } u_i^{(n)} = u_{i,3}^{(n)} \text{ with } q_{\dot{q}i}^{(k)}(n,3), \ldots, \ u_i^{(n)} = u_{i,p}^{(n)} \text{ with} \tag{35}$$
$$q_{\dot{q}i}^{(k)}(n,p), \ldots, u_i^{(n)} = u_{i,P}^{(n)}. \text{ with } q_{\dot{q}i}^{(k)}(n,P)$$

where $\mathfrak{U}_{i,n} = \left\{ u_{i,1}^{(n)}, ..u_{i,p}^{(n)}, .., u_{i,P}^{(n)} \right\}$ is the set of action candidates in the rule $R_{n,i}$. $L_i^{(n)} = \{ L_{1,i}^{(n)}, \ldots, L_{m,i}^{(n)} \}$ is the set of linguistic variables of fuzzy inputs. $s_i^{(k)} = \left\{ s_{1,i}^{(k)}, \ldots, s_{m,i}^{(k)} \right\}$ is the set of fuzzy inputs at the $k$ moment. In this paper, the fuzzy inputs are $e_i$ and $\dot{q}_i$ such that $s_i^{(k)} = \left\{ e_i(k), \dot{q}_i(k) \right\}$.

The set of fuzzy inputs $s_i^{(k)} = \left\{ s_{1,i}^{(k)}, \ldots, s_{m,i}^{(k)} \right\}$ is fuzzified by the membership function shown in Figure 2 and then matched with the rule antecedents (19), providing the firing rate vector $\varphi\left( s_i^{(k)} \right) = \left[ \varphi_1\left( s_i^{(k)} \right), \varphi_2\left( s_i^{(k)} \right), .., \varphi_n\left( s_i^{(k)} \right) \right]$. $n$ is the amount of fuzzy rules (There are $n$ fuzzy rules).

For the $n^{th}$ rule in the $i^{th}$ subsystem ($R_{n,i}$), the optimal action at the $k$ moment is defined as the action with the maximum $q_{\dot{q}i}^{(k)}(n,p)$, $p \in \{1, 2, .., P\}$ among $P$ action candidates.

$$u_i^{*(n)} = arg \max_{u_i^{(n)} \in \mathfrak{U}_{i,n}} q_{\dot{q}i}^{(k)}\left( n, u_{i,p}^{(n)} \right) \tag{36}$$

To prevent the selection of $\mathrm{u}$ from the local optimum in the learning process, we introduce a greed mechanism:

$$\hat{\mathrm{u}}_i^{(n)} = \begin{cases} \mathrm{u}^{+}{}_i^{(n)}, \text{ with probablity } \rho \\ \mathrm{u}^{*}{}_i^{(n)} \text{ with probablity } 1-\rho \end{cases} \tag{37}$$

where $\mathrm{u}^{+}{}_i^{(n)}$ is an action randomly selected from $\mathfrak{U}_{i,n}$. $0 < \rho < 1$ is the probability to explore random actions.

The numerical value of $\Lambda_i$ is calculated by firing rates and the selected actions:

$$\Lambda_i\left(s_i^{(\Bbbk)}\right) = \frac{\sum_{n=1}^{\mathcal{N}} \varphi n\left(s_i^{(\Bbbk)}\right)\hat{\mathrm{u}}_i^{(n)}}{\sum_{n=1}^{\mathcal{N}} \varphi n\left(s_i^{(\Bbbk)}\right)} \tag{38}$$

The update principle of $q$ is the most important part in the whole learning process. The $q$-values are updated according to the rewards of the selected actions; the optimal action can achieve higher rewards, therefore, we can finally learn the optimal action.

The Q value at the $\Bbbk$ moment can be designed as follows:

$$Q\left(s_i^{(\Bbbk)}\right) = \frac{\sum_{n=1}^{\mathcal{N}} \varphi n\left(s_i^{(\Bbbk)}\right)q_i^{(\Bbbk)}\left(n,\,\hat{\mathrm{u}}_i^{(n)}\right)}{\sum_{n=1}^{\mathcal{N}} \varphi n\left(s_i^{(\Bbbk)}\right)} \tag{39}$$

The target value at the state $s_i^{(k)}$ is calculated as:

$$\mathcal{V}\left(s_i^{(\Bbbk)}\right) = \frac{\sum_{n=1}^{\mathcal{N}} \varphi n\left(s_i^{(\Bbbk)}\right)q_i^{(\Bbbk)}\left(n,\,\mathrm{u}^{*}{}_i^{(n)}\right)}{\sum_{n=1}^{\mathcal{N}} \varphi n\left(s_i^{(\Bbbk)}\right)} \tag{40}$$

When the system state $s_i^{(\Bbbk)}$ is driven to the next state $s_i^{(\Bbbk+1)}$, temporal difference (TD) is calculated according to the reward obtained at the $k$ moment $r_i^{(\Bbbk)}$:

$$\Delta Q_i^{(\Bbbk)} = r_i^{(\Bbbk)} + \alpha_i \mathcal{V}\left(s_i^{(\Bbbk+1)}\right) - Q\left(s_i^{(\Bbbk)}\right) \tag{41}$$

where $\alpha_i \in [0,1]$ is the discount factor reflecting the contribution of the future reward. The reward $r_i^{\Bbbk}$ at the $\Bbbk$ moment of the $i^{th}$ subsystem is designed in (42) and the meaning of (42) is explained in Remark 9.

$$r_i^{(\Bbbk)} = \begin{cases} 0,\, if\, |\S_i(\Bbbk)| > \sigma_i \\ \cos\left(\frac{\pi}{2}\frac{|\S_i(\Bbbk)|}{\sigma_i}\right),\, if\, |\S_i(\Bbbk)| \le \sigma_i \end{cases} \tag{42}$$

where $\S_i(\Bbbk) = \dot{q}_i(\Bbbk) + \Lambda_i tanh\left(\frac{e_i(\Bbbk)}{\mathfrak{C}_i}\right)$ and $\sigma_i > 0$ is a positive constant.

Finally, the adaptive law of q-values is:

$$q_i^{(\Bbbk+1)}\left(n,\hat{\mathrm{u}}_i^{(n)}\right) = q_i^{(\Bbbk)}\left(n,\hat{\mathrm{u}}_i^{(n)}\right) + \Gamma_i \cdot \Delta Q_i^{(\Bbbk)} \cdot \frac{\varphi n\left(s_i^{(\Bbbk)}\right)}{\sum_{n=1}^{\mathcal{N}} \varphi n\left(s_i^{(\Bbbk)}\right)} \tag{43}$$

where $\Gamma_i \in [0, 1]$ is the learning rate.

**Remark 9.** *The reward function (42) indicates that higher values of reward $r_i^{(\Bbbk)}$ can be obtained by the smaller $\S_i(\Bbbk)$, which means the the action that satisfies (24) will be given with the highest reward ($r_i^{(\Bbbk)} = 1$).*

The aforementioned process of fuzzy Q learning to tune $\Lambda_i$ can be also concluded in Figure 3. The proposed control scheme can be concluded in Figure 4.

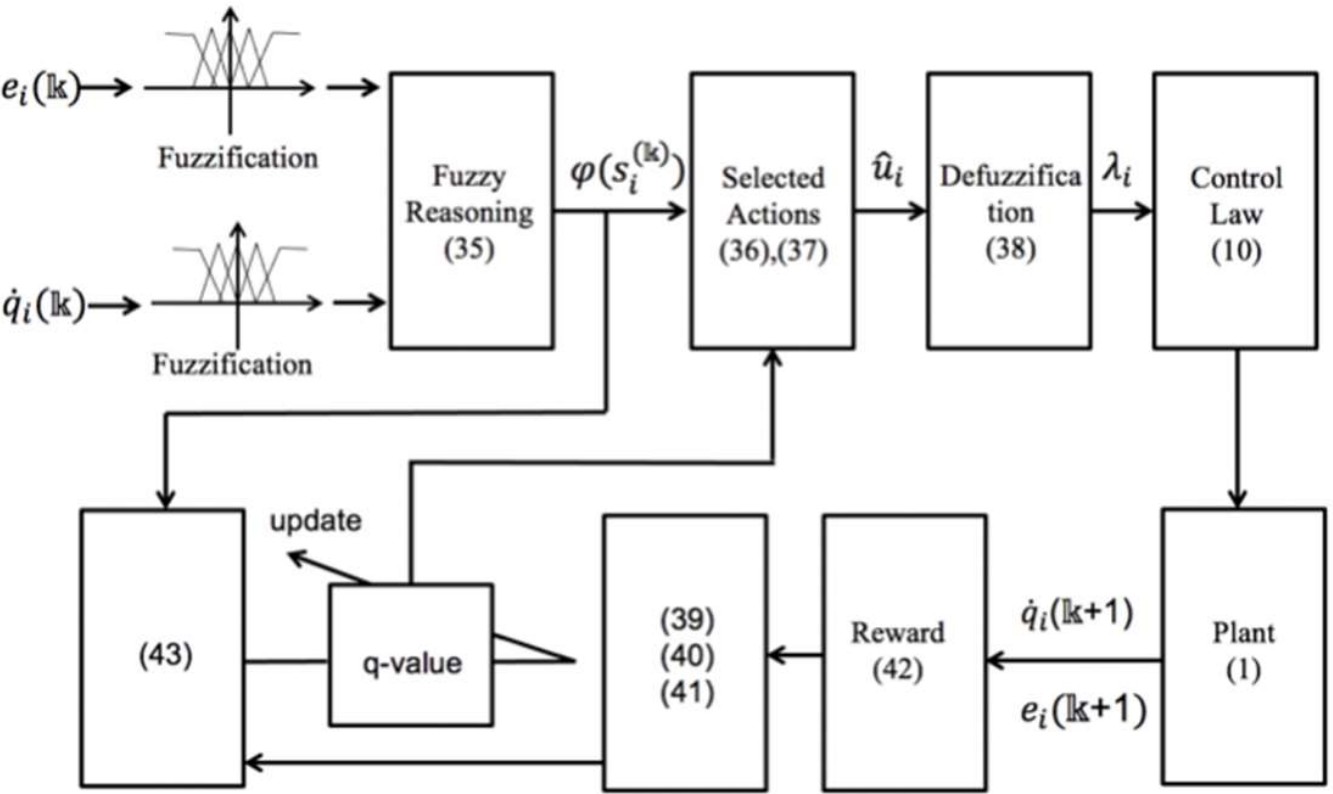

**Figure 3.** Flow chart of fuzzy Q learning to determine $\Lambda_i$.

**Remark 10.** *All the system states including the tracking errors are proven to be bounded with $\Lambda_i > 0$. Therefore, the instability will not occur even if an inappropriate positive value of $\Lambda_i$ is calculated by the fuzzy Q learning mechanism trying some bad action candidates. Hence, the designed control law (7), (9)–(14) offers the fuzzy Q learning mechanism (32)–(43) a safe environment to learn a optimal $\Lambda_i$. Notably, in order to make sure $\Lambda_i > 0$ always holds, all the action candidates $\mathfrak{U}_{i,n} = \left\{ u_{i,1}^{(n)}, ..u_{i,p}^{(n)}, .., u_{i,P}^{(n)} \right\}$ should be positive.*

**Remark 11.** *The proof of Theorem 1 requires the parameter $\Lambda_i > 0$ to be a constant. However, the parameter $\Lambda_i$ is oneline tuned by a fuzzy Q learning mechanism, and therefore $\Lambda_i$ is varying in the implementation of the proposed controller. To handle this issue, we design that the time interval between two consecutive fuzzy inferences of $\Lambda_i$ in fuzzy Q learning is 20-times greater than the time interval between two consecutive control actions $\tau$ (eg. 0.1s between two consecutive $\hat{u}_i^{(n)}$ (equally, two consecutive $\Lambda_i$) while 0.001s between two consecutive $\tau$). Namely, $\Lambda_i(t) = \Lambda_i(t^*)$, $\forall t \in [t^*, t^* + 20L]$; $\tau(t) = \tau(t^*)$, $\forall t \in [t^*, t^* + L]$. $t^*$ is any time moment when the algorithm works. Thereby, not only can the time derivative of $\Lambda_i$ be negligible in the proof of Theorem 1, but the computational load is also decreased.*

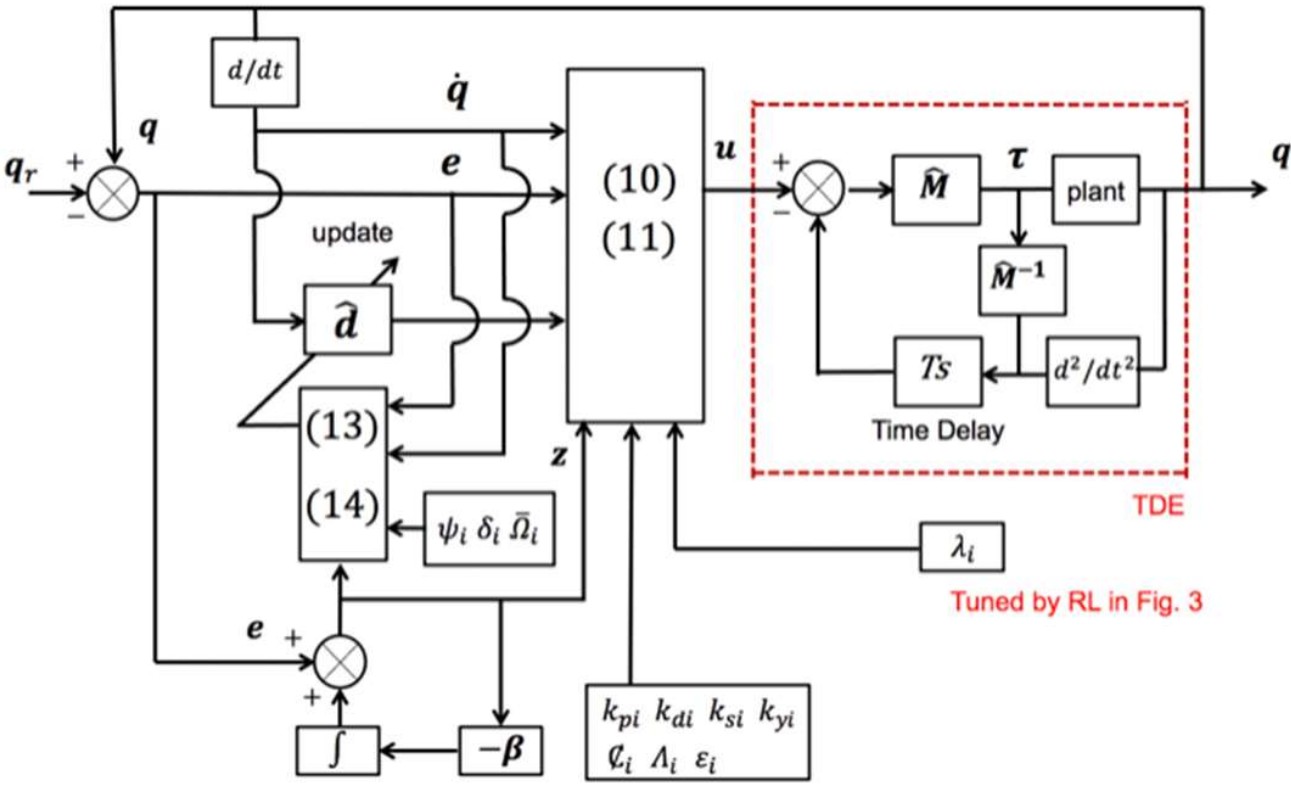

**Figure 4.** Proposed RL-based adaptive control scheme.

## 4. Simulation Results and Analysis

In this section, similar to works [38–42], we use the simulation to verify the effectiveness of the proposed controller. A 2-rigid-link robotic manipulator shown in Figure 5 is carried in Matlab 2018a. The sampling time of simulating the real dynamics of robotic manipulators is set as $1 \times 10^{-4}$ s. The sampling time of TDE and controller is set as $1 \times 10^{-3}$ s (10 times to $1 \times 10^{-4}$) to show the discrete nature of using controllers in practice. The sampling time of fuzzy Q learning is set as 0.01 s according to Remark 10. The dynamic model of a 2-rigid-link robotic manipulator is given as follows, which can be also found in [26].

$$M(q) = \begin{bmatrix} m_2 l_2^2 + 2l_1 l_2 m_2 \cos(q_2) + (m_1 + m_2) l_1^2 & m_2 l_2^2 + l_1 l_2 m_2 \cos(q_2) \\ m_2 l_2^2 + l_1 l_2 m_2 \cos(q_2) & m_2 l_2^2 \end{bmatrix}$$

$$C(q, \dot{q})\dot{q} = \begin{bmatrix} -m_2 l_1 l_2 \sin(q_2)\dot{q}_2^2 - 2m_2 l_1 l_2 \sin(q_2)\dot{q}_1\dot{q}_2 \\ m_2 l_1 l_2 \sin(q_2)\dot{q}_1^2 \end{bmatrix}$$

$$G(q) = \begin{bmatrix} (m_1 + m_2) l_1 cos(q_2)g + m_2 l_2 cos(q_1 + q_2)g \\ m_2 l_2 \cos(q_1 + q_2)g \end{bmatrix}$$

$$F(\dot{q}) = \begin{bmatrix} F_1 \\ F_2 \end{bmatrix}$$

$$\tau_d = \begin{bmatrix} \tau_{d1} \\ \tau_{d2} \end{bmatrix}$$

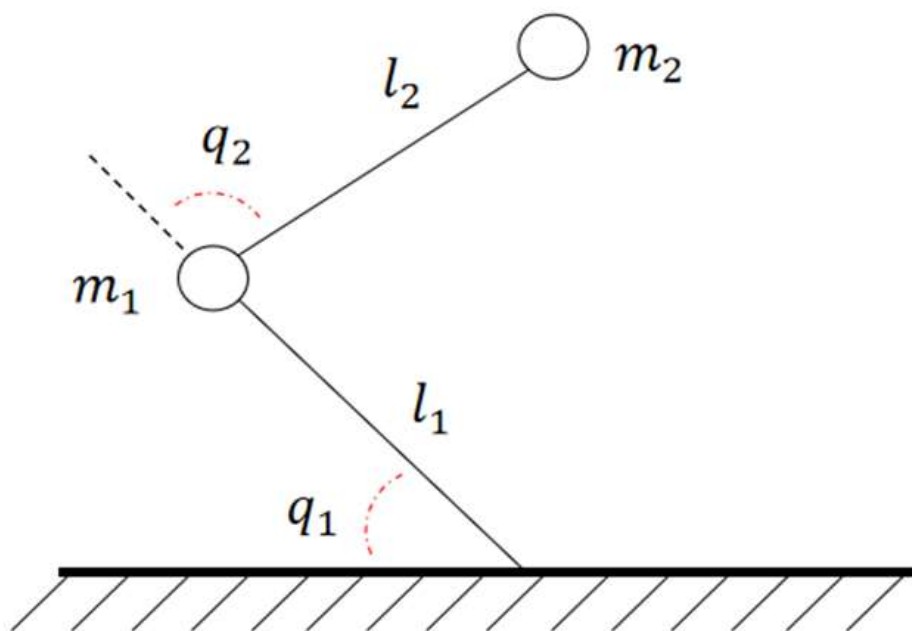

**Figure 5.** 2-rigid-link robotic manipulator.

The system parameters are given in Table 1, which are the same as [26].

**Table 1.** Parameters of robotic manipulator.

|  | Parameter | Value |
|---|---|---|
|  | $l_1$ (*m*) | 0.5 |
|  | $l_2$ (*m*) | 0.5 |
| Robotic manipulator | $m_1$ (*kg*) | 5 |
|  | $m_2$ (*kg*) | 2 |
|  | $g$ (*m/s$^2$*) | 9.8 |

The initial angular positions and velocities of two joints are set as: $q_1 = 0^o$, $q_2 = 0^o$, $\dot{q}_1 = 0^o/s$ and $\dot{q}_2 = 0^o/s$. The desired angular positions of two joints are set as: $q_{r1} = 30^o$ and $q_{r2} = 45^o$.

The upper bounds of the angular position and angular velocity for the $i^{th}$ joint of manipulator ($\varepsilon_i$ and $\Lambda_i$) are selected by the user dependent on the specific mission requirement. To successfully implement the proposed controller, users can select any values satisfying $\varepsilon_i > |q_{ri}|$ and $\Lambda_i > 0$. Therefore, in the simulation of this paper, we select $\varepsilon_1 = 50^o$, $\varepsilon_2 = 60^o$, $\Lambda_1 = 10^o/s$ and $\Lambda_2 = 12^o/s$.

**Remark 12.** *$k_{pi}$ and $k_{di}$ are the proportional coefficient and differential coefficient, respectively. The great $k_{pi}$ could decrease steady state error, but an excessively great could result in a significant overshoot. While the great $k_{di}$ could improve the robustness to disturbance/uncertainty, but an over-great $k_{di}$ could compromise the tracking accuracy. The selection of $k_{pi}$ and $k_{di}$ can be based on the tuning rules of PID controllers.*

**Remark 13.** *Great $k_{si}$ can amplify the effect of the terms $A_i$ and $B_i$ in (10) to keep the angular positions from hitting their constraints. An inappropriately small $k_{si}$ could lead to a $|q_i|$ too close to $\varepsilon_i$ and then result in an over-great $u_i$ in (10). Therefore, the selection of $k_{si}$ could be started at a small value, and then users could gradually increase $k_{si}$ until the magnitude of $u_i$ is acceptable. Great $k_{yi}$ can amplify the effect of the term $C_i$ in (10) to keep the angular velocities from hitting their constraints. An excessively great $k_{yi}$ could lead to an over great $(1 + k_{yi}C_i)$ in (10) to decline the converging rate of the tracking error even if the angular velocity is not close to its constraint, which*

could increase the settling time. Therefore, the trial of selecting $k_{yi}$ could start at a small value, and then users could decrease it until the converging rate of the tracking error is acceptable. $\beta_i$ determines the converging rate of the auxiliary variable $z_i$ that has a significant effect on steady state error in the controller [33]. However, in the proposed controller, the RL-based term is introduced to improve the tracking performance and thereby the effect of $\beta_i$ is decreased. We suggest to offer $\beta_i$ with a medium value ranging between 0.1–1.

**Remark 14.** *large values of* $\psi_i$ *and* $\delta_i$ *can lead to a fast adaption of* $\hat{d}_i$ *to handle system uncertainty and external disturbance. However, the inappropriately large* $\psi_i$ *and* $\delta_i$ *could result in a sharp variation of* $\hat{d}_i$ *and then a chattering effect on control torques. Therefore, the selection of* $\psi_i$ *and* $\delta_i$ *could start at a big value, and then users could decrease them until no chattering effect occurs on the control torques. Over-small values of* $\overline{\Omega}_i$ *could lead to an insufficient decrease of* $\hat{d}_i$ *that still brings up a chattering effect on control torques. Meanwhile, over-great values of* $\overline{\Omega}_i$ *could negatively influence the robustness to uncertainty and disturbance. Hence, the selection of* $\overline{\Omega}_i$ *could start at some small values, and then users could increase* $\overline{\Omega}_i$ *until* $\hat{d}_i$ *is significantly decreased at the final stage of the control to have a satisfactory chattering attenuation.*

**Remark 15.** *Small/great values of* $\mathbb{C}_i$ *can amplify/reduce the effect of the RL term* $\Lambda_i \Lambda_i tanh\left(\dfrac{e_i}{\mathbb{C}_i}\right)$ *in (10). The over-small values of* $\mathbb{C}_i$ *could result in the over-great magnitude of* $u_i$*, while the over-great values of* $\mathbb{C}_i$ *could bring up an insufficient improvement on tracking performance. Therefore, the trial of selecting* $\mathbb{C}_i$ *is suggested to start at a great value, and then users could decrease* $\mathbb{C}_i$ *until a satisfying improvement on the tracking performance.*

According to Remark 11~Remark 15, the parameters of the proposed controller are selected in Table 2.

**Table 2.** Parameters of proposed controller.

| Parameter | Value (*i*=1) | Value (*i*=2) |
|---|---|---|
| $k_{pi}$ | 10 | 10 |
| $k_{di}$ | 1 | 1 |
| $k_{si}$ | 0.1 | 0.1 |
| $\mathbb{C}_i$ | 0.1 | 0.1 |
| $k_{yi}$ | 0.1 | 0.1 |
| $\beta_i$ | 0.1 | 0.1 |
| $\psi_i$ | 0.01 | 0.01 |
| $\overline{\Omega}_i$ | 0.05 | 0.05 |
| $\delta_i$ | 0.1 | 0.1 |

Notably, it does not require an extensive trial to select parameters in Table 2 as the satisfying tracking performance is mainly obtained by the optimal $\Lambda_i$ in the RL term $\Lambda_i \Lambda_i tanh\left(\dfrac{e_i}{\mathbb{C}_i}\right)$. In other words, users can select some medium values for parameters (not optimal) in Table 2 by an acceptable amount of parameters-selection trails. Then, the fuzzy Q learning can automatically explore the $\Lambda_i$ matching the selected parameters to have a satisfying tracking performance.

**Remark 16.** *Similar to [36], the parameters of membership function in (33) and (34), $Lin(e_i) = \left\{ \zeta_{1,1}^{(i)}, \zeta_{1,2}^{(i)}, \ldots, \zeta_{1,a}^{(i)} \right\}$ and $Lin(\dot{q}_i) = \left\{ \zeta_{2,1}^{(i)}, \zeta_{2,2}^{(i)}, \ldots, \zeta_{2,b}^{(i)} \right\}$, are used to fuzzify $e_i$ and $\dot{q}_i$. To well present the $e_i$ and $\dot{q}_i$ in the form of firing rate, we suggest to offer $\zeta_{1,1}^{(i)}$ and $\zeta_{2,1}^{(i)}$ with some small values and provide some great values to $\zeta_{1,a}^{(i)}$ and $\zeta_{2,b}^{(i)}$. The $\zeta_{1,a}^{(i)}$ and $\zeta_{2,b}^{(i)}$ are suggested to be evenly distributed among $(\zeta_{1,1}^{(i)}, \zeta_{1,a}^{(i)})$ and $(\zeta_{2,1}^{(i)}, \zeta_{2,b}^{(i)})$, respectively. The great $a$ and $b$ increase the potential to well present $e_i$ and $\dot{q}_i$ at the cost of increasing the computational load. Therefore, the*

*trial of selecting $a$ and $b$ could start at some great values (e.g., 20), and then users could decrease them until a reasonable computational load.*

**Remark 17.** *$\sigma_i$ is the threshold of obtaining rewards. The inappropriately small values of $\sigma_i$ could result in a difficulty in obtaining high values of reward, while the over-great values of $\sigma_i$ could result in the different action candidates to be offered with the similarly high rewards even if they lead to the different control performances. Therefore, both of the over-great and over-small values of $\sigma_i$ will negatively influence the convergence of q-values in (43) and thereby compromise the performance of reinforcement learning. The selection of $\sigma_i$ could start at a small value, and the users could increase $\sigma_i$ until a satisfying convergence of q values (the convergence of q-values can be also reflected by the convergence of obtained rewards). The selection of mutation probability $\rho$, learning rate $\Gamma_i$ and discount factor $\alpha_2$ can be based on the strategy mentioned in [36].*

**Remark 18** *Action candidates $\mathfrak{U}_{i,n} = \left\{ u_{i,1}^{(n)}, .. u_{i,p}^{(n)}, .., u_{i,P}^{(n)} \right\}$ are the most important parameters in the proposed control scheme because the optimal $\Lambda_i$ that brings up a satisfying control performance is calculated based on them. To make the optimal action included in the group of action candidates, $u_{i,1}^{(n)}$ should be given a small value (e.g., 0) while $u_{i,P}^{(n)}$ should be given a great value. The rest candidates $u_{i,p}^{(n)}$ ($p = 1, 2, .., p$) are suggested to be evenly distributed between $u_{i,1}^{(n)}$ and $u_{i,P}^{(n)}$. Users could initially give $u_{i,P}^{(n)}$ with a small value (but greater than $u_{i,1}^{(n)}$), and an increase of $u_{i,P}^{(n)}$ until a sufficient improvement on the tracking performance is achieved. $p$ is the amount of action candidates of each fuzzy rule for each subsystem. The selection of $p$ could start at a great number (e.g., 20), and then users could decrease $p$ until the computational load is acceptable.*

According to *Remark 16~Remark 18*, the parameters of the fuzzy Q learning mechanism are given as follows. The amount of fuzzy sets in (17) are $a = b = 8$. Therefore, the amount of fuzzy rules are $n = 64$. The parameters of membership function to do the fuzzification are: $Lin(e_1) = \left\{ \zeta_{1,1}^{(1)}, \zeta_{1,2}^{(1)}, \dots, \zeta_{1,8}^{(1)} \right\} = \{-0.0017, -0.0014, -0.001, -0.0006, -0.0002, 0.0002, 0.0006, 0.001, 0.0014, 0.0017\}$. $Lin(\dot{q}_1) = \left\{ \zeta_{2,1}^{(1)}, \zeta_{2,2}^{(1)}, \dots, \zeta_{2,8}^{(1)} \right\} = \{-0.175, -0.138, -0.097, -0.058, -0.019, 0.019, 0.058, 0.097, 0.138, 0.175\} \times 10^{-3}$. $Lin(e_2) = \left\{ \zeta_{1,1}^{(2)}, \zeta_{1,2}^{(2)}, \dots, \zeta_{1,8}^{(2)} \right\} = \{-0.0017, -0.0014, -0.001, -0.0006, -0.0002, 0.0002, 0.0006, 0.001, 0.0014, 0.0017\}$. $Lin(\dot{q}_2) = \left\{ \zeta_{2,1}^{(2)}, \zeta_{2,2}^{(2)}, \dots, \zeta_{2,8}^{(2)} \right\} = \{-0.175, -0.138, -0.097, -0.058, -0.019, 0.019, 0.058, 0.097, 0.138, 0.175\} \times 10^{-3}$. The action candidates for each fuzzy rule are: $\mathfrak{U}_{i,n} = \{0, 22, 44, 66, 88, 111, 133, 155, 177, 200\}$ for all $i = 1, 2$ and $n = 1, 2, \dots, 64$. Therefore, the amount of action candidates for each fuzzy rule is $P = 10$. The discount factor is $\alpha_1 = \alpha_2 = 0.01$ and the learning rate is $\Gamma_1 = \Gamma_2 = 0.2$. The threshold of obtaining rewards is $\sigma_1 = \sigma_2 = 0.01$.

To show the superiority, four existing controllers, Refs. [26,33,43,44] are compared with the proposed controller.

The controller from [26] is given as (44)–(52)

$$\tau = \tau_1 + \tau_2 \tag{44}$$

$$\tau_1 = \hat{M}(q)\left(\ddot{q}_r - K_V \dot{e} - K_P e\right) + \hat{C}(q, \dot{q})\dot{q} + \hat{G}(q) \tag{45}$$

$$\tau_2 = -\hat{M}(q)\hat{f} + \hat{M}(q)u_r \tag{46}$$

$$u_r = \begin{bmatrix} -\hat{\xi}\tan h(a_1 p/\rho_1) \\ -\hat{\xi}\tan h(a_2 p/\rho_1) \end{bmatrix} \tag{47}$$

$$\hat{f} = \hat{W}^T \sigma\left(\hat{V}^T X\right) \tag{48}$$

$$\dot{\hat{\xi}} = a_1 p \tanh\left(\frac{a_1 p}{\rho_1}\right) + a_1 p \tanh\left(\frac{a_2 p}{\rho_1}\right) - K\hat{\xi} \tag{49}$$

$$\dot{\hat{W}} = \left(\sigma - \sigma'\hat{V}^T X\right) x^T PB - Y_W \hat{W} \tag{50}$$

$$\dot{\hat{V}} = X x^T PB \hat{W}^T \sigma' - Y_V \hat{V} \tag{51}$$

$$p = 1 + ||X|| + ||\hat{V}|| \cdot ||X|| + ||\hat{W}|| \cdot ||X|| \tag{52}$$

where $\sigma = [\sigma_1, \sigma_2, \ldots, \sigma_m]^T$ is the vector of hidden neurons with the activation function $\sigma_i(s) = 1/(1 + e^{-s})$. $\sigma'$ is the vector of partial derivative of $\sigma$ such that $\sigma' = \partial\sigma(s)/\partial s$. $x = [e_1, e_2, \dot{q}_1, \dot{q}_2]^T$ and $X = [q_{r1}, q_{r2}, q_1, q_2, \dot{q}_1, \dot{q}_2, \ddot{q}_1, \ddot{q}_2]^T$. $P$ satisfies $PA + A^T P = -Q$ with $A = \begin{bmatrix} 0 & I \\ -K_P & -K_V \end{bmatrix}$. $[a_1, a_2] = x^T PB$.

In this simulation, we let the controller from [26] to fully know the system parameters such that $\hat{M}(q) = M(q)$, $\hat{C}(q, \dot{q}) = C(q, \dot{q})$ and $\hat{G}(q) = G(q)$.

The parameters in (44)–(52) are given as:

$$K_V = \begin{bmatrix} 300 & 0 \\ 0 & 300 \end{bmatrix}, \ K_P = \begin{bmatrix} 200 & 0 \\ 0 & 200 \end{bmatrix}$$

$$B = \begin{bmatrix} 0 & 0 \\ 0 & 0 \\ 1 & 0 \\ 0 & 1 \end{bmatrix}, \ Q = \begin{bmatrix} 1 & 0 & 0 & 0 \\ 0 & 1 & 0 & 0 \\ 0 & 0 & 1 & 0 \\ 0 & 0 & 0 & 1 \end{bmatrix}$$

$\rho_1 = 0.01$, $K = 0.005$, $Y_W = 0.15$, $Y_V = 0.15$, $\hat{\xi}(0) = 0.01$

$$\hat{W}(0) = \begin{bmatrix} -0.1 & -0.1 \\ -0.1 & -0.1 \\ -0.1 & -0.1 \\ -0.1 & -0.1 \\ -0.1 & -0.1 \end{bmatrix}$$

$$\hat{V}(0) = \begin{bmatrix} 0.1 & 0.1 & 0.1 & 0.1 & 0.1 \\ 0.1 & 0.1 & 0.1 & 0.1 & 0.1 \\ 0.1 & 0.1 & 0.1 & 0.1 & 0.1 \\ 0.1 & 0.1 & 0.1 & 0.1 & 0.1 \\ 0.1 & 0.1 & 0.1 & 0.1 & 0.1 \\ 0.1 & 0.1 & 0.1 & 0.1 & 0.1 \end{bmatrix}$$

The controller from [33] is given as (53)–(58).

$$\tau = \mathcal{G}(q)\hat{p} - K_d\dot{q} - K_p e - K_s diag\{\mathcal{X}_1, \mathcal{X}_2\}z \tag{53}$$

$$\mathcal{X}_i = \frac{\zeta_i^2}{\zeta_i^2 - q_i^2} + \frac{\zeta_i^2 q_i z_i}{\left(\zeta_i^2 - q_i^2\right)^2}, \ i = 1, 2 \tag{54}$$

$$z_i = e_i + \int_0^t \eta_i(\Theta)d\Theta, \ i = 1, 2 \tag{55}$$

$$\eta_i = -\beta_i z_i, \ i = 1, 2 \tag{56}$$

$$\dot{\hat{p}} = -\Psi \mathcal{G}^T(q)\dot{q} \tag{57}$$

$$\mathcal{G}(q) = \begin{bmatrix} \cos(q_1 + q_2) & \cos(q_2) \\ \cos(q_1 + q_2) & 0 \end{bmatrix} \tag{58}$$

where $\hat{p} = [\hat{p}_1, \hat{p}_2]^T$, $z_1(0) = e_1(0)$ and $z_2(0) = e_2(0)$. $\zeta_1$ and $\zeta_2$ are the specific constraint (designed upper bound) of the 1st and 2nd joint respectively, therefore, $\zeta_1 = \varepsilon_1 = 50^o$ and $\zeta_2 = \varepsilon_2 = 60^o$.

The parameters in (53)–(58) are given as follows:

$$K_d = \begin{bmatrix} 60 & 0 \\ 0 & 20 \end{bmatrix}, \ K_p = \begin{bmatrix} 24 & 0 \\ 0 & 24 \end{bmatrix}, \ K_s = \begin{bmatrix} 20 & 0 \\ 0 & 20 \end{bmatrix}$$

$$\beta_1 = 30, \ \beta_2 = 30, \ \Psi = \begin{bmatrix} 0.1 & 0 \\ 0 & 0.1 \end{bmatrix}$$

The controller from [43] is given as (59)–(61).

$$\tau = K_D^{-1}\hat{M}\left[K_P\dot{e} - K_I e + K_D\ddot{q}_r + \beta\dot{s} - K_D\hat{\Gamma} - K_D\Lambda s - K_D k_s sgn(\dot{s})\right] \tag{59}$$

$$\dot{s} + \beta s = K_P e + K_I \int_0^t e + K_D\dot{e} \tag{60}$$

$$\dot{k}_s = \Gamma||\dot{s}|| \tag{61}$$

where $\dot{s}(0) = [0,0]^T$, $k_s(0) = 0$, $\Gamma = 0.1$, $\hat{\Gamma}$ is the known part of lumped uncertainty $\Gamma$. In this simulation, we let $\hat{\Gamma} = 0.7\Gamma$. The parameters $K_P$, $K_D$, $K_I$ and $\beta$ are given as follows.

$$K_D = \begin{bmatrix} 20 & 0 \\ 0 & 20 \end{bmatrix}, \ K_P = \begin{bmatrix} 10 & 0 \\ 0 & 10 \end{bmatrix}, \ K_I = \begin{bmatrix} 0.01 & 0 \\ 0 & 0.01 \end{bmatrix}, \ \beta = \begin{bmatrix} 0.1 & 0 \\ 0 & 0.1 \end{bmatrix}$$

The controller from [44] is given as (62)–(65).

$$\begin{cases} \hat{D} = z + Y\dot{q} \\ \dot{z} = -Y\hat{M}^{-1}z + Y\hat{M}^{-1}\left(-\tau - Y\dot{q} - \overline{D}\right) \end{cases} \tag{62}$$

$$\tau = \hat{M}(\ddot{q}_r - \Lambda\dot{q} + \Lambda\dot{q}_r) - \overline{D} - K_D sat(s) - \hat{D} \tag{63}$$

$$sat(s_i) = \begin{cases} sgn(s_i), \ |s_i| \geq \sigma \\ s_i/\sigma, |s_i| < \sigma \end{cases} \tag{64}$$

where $\sigma = 0.1$, $sat(s) = [s_1, s_2]^T$. $z(0) = [0,0]^T$ and $s = \dot{e} + \Lambda e$. $\overline{D}$ is known part of $\hat{M}^{-1}\Gamma$, $\overline{D} = 0.7\hat{M}^{-1}\Gamma$ in the simulation. The parameters $Y$, $\Lambda$ and $K_D$ are given as follows.

$$K_D = \begin{bmatrix} 40 & 0 \\ 0 & 16 \end{bmatrix}, \ Y = \begin{bmatrix} \frac{1}{0.06} & 0 \\ 0 & \frac{1}{0.06} \end{bmatrix}, \ \Lambda = \begin{bmatrix} 40 & 0 \\ 0 & 20 \end{bmatrix}$$

Furthermore, two cases are considered to run the simulation. In the 1st case, the parameters of the dynamic model are fully known, and there is no external disturbance and no friction. In the 2nd case, the parametric uncertainty is considered and the unknown external disturbance and frictions are applied on the dynamics model.

**Remark 19.** *The selections of parameters for the proposed controller, the controller from [26] and the controller from [33] are all carried out in case 1. In other words, all the parameters for the three different controllers are fine-tuned to have a good performance in case 1. In case 2, all the selected parameters of the three controllers remain unchanged to test the robustness to the lumped uncertainty.*

**Case 1. In the absence of system uncertainty, external disturbance and friction.**

In this case, the system parameters used in the three controllers (proposed controller, controller from [26] and controller from [33]) are the same as the parameters in the dynamics model of manipulator, which means no parametric uncertainty. Meanwhile, the friction and external disturbance are null, shown as follows:

$$F(\dot{q}) = \begin{bmatrix} F_1 \\ F_2 \end{bmatrix} = \begin{bmatrix} 0 \\ 0 \end{bmatrix}, \ \tau_d = \begin{bmatrix} \tau_{d1} \\ \tau_{d2} \end{bmatrix} = \begin{bmatrix} 0 \\ 0 \end{bmatrix}$$

The comparisons of angular position tracking are shown in Figures 6 and 7, while the comparisons of angular velocity are shown in Figures 8 and 9. Cleanly, in the absence of unknown disturbance and system parametric uncertainty, the proposed controller shows an inferior performance in terms of a greater steady state error and a slower error converging rate, compared to the controllers from [26,33]. In comparison to [43,44], it provides a response faster than the sliding mode controller [44] but slower than the disturbance observer-based controller [44]. However, the steady state error of the proposed controller is smaller than that of [43,44].

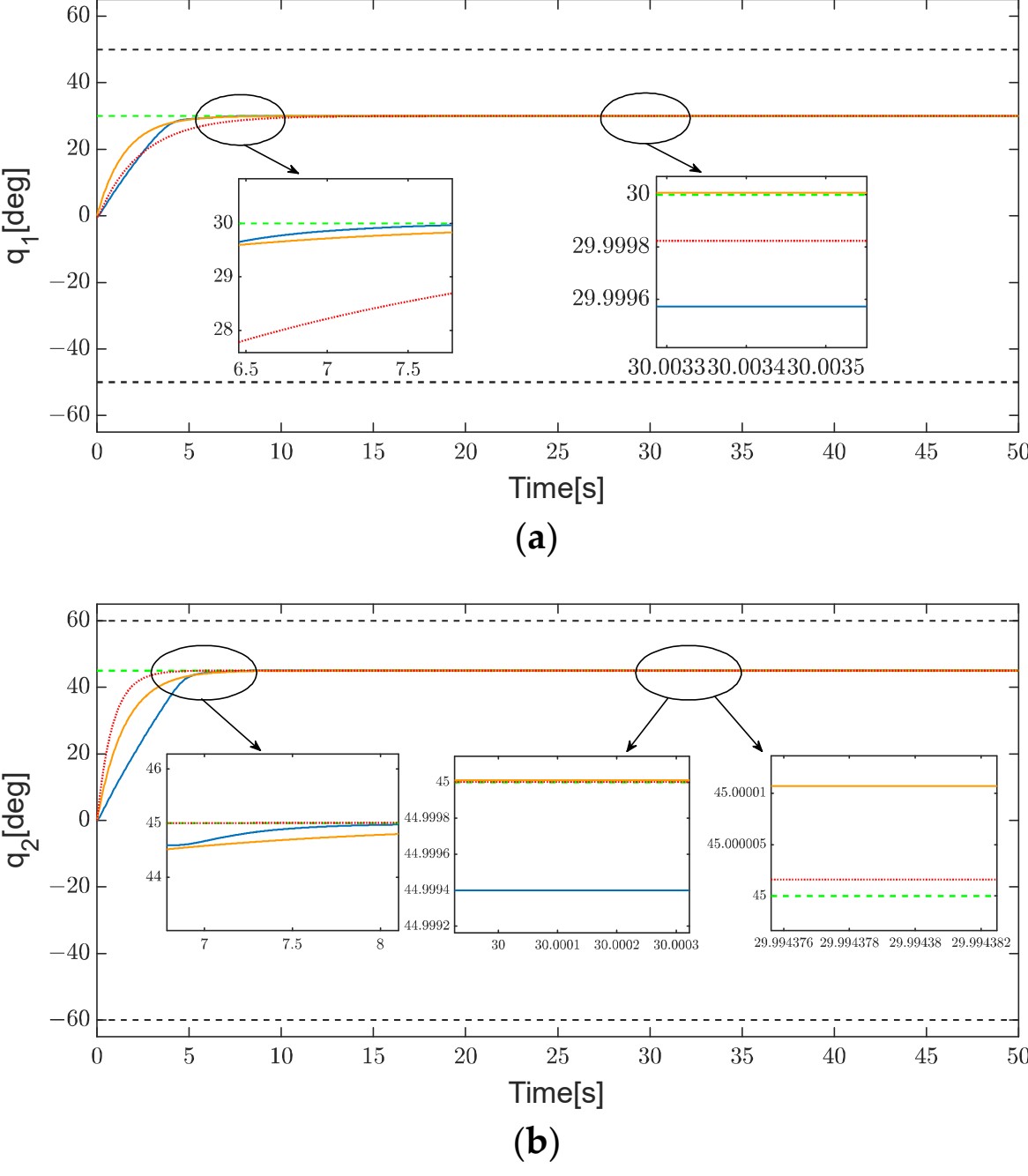

**Figure 6.** Comparison of angular position to [26,33] in case 1: [26] scheme (yellow solid line), [33] scheme (red dash line), the proposed scheme (blue solid line), the reference angular position (green dashed line), the angular position constrains (black dashed line); (**a**) Joint 1. (**b**) Joint 2.



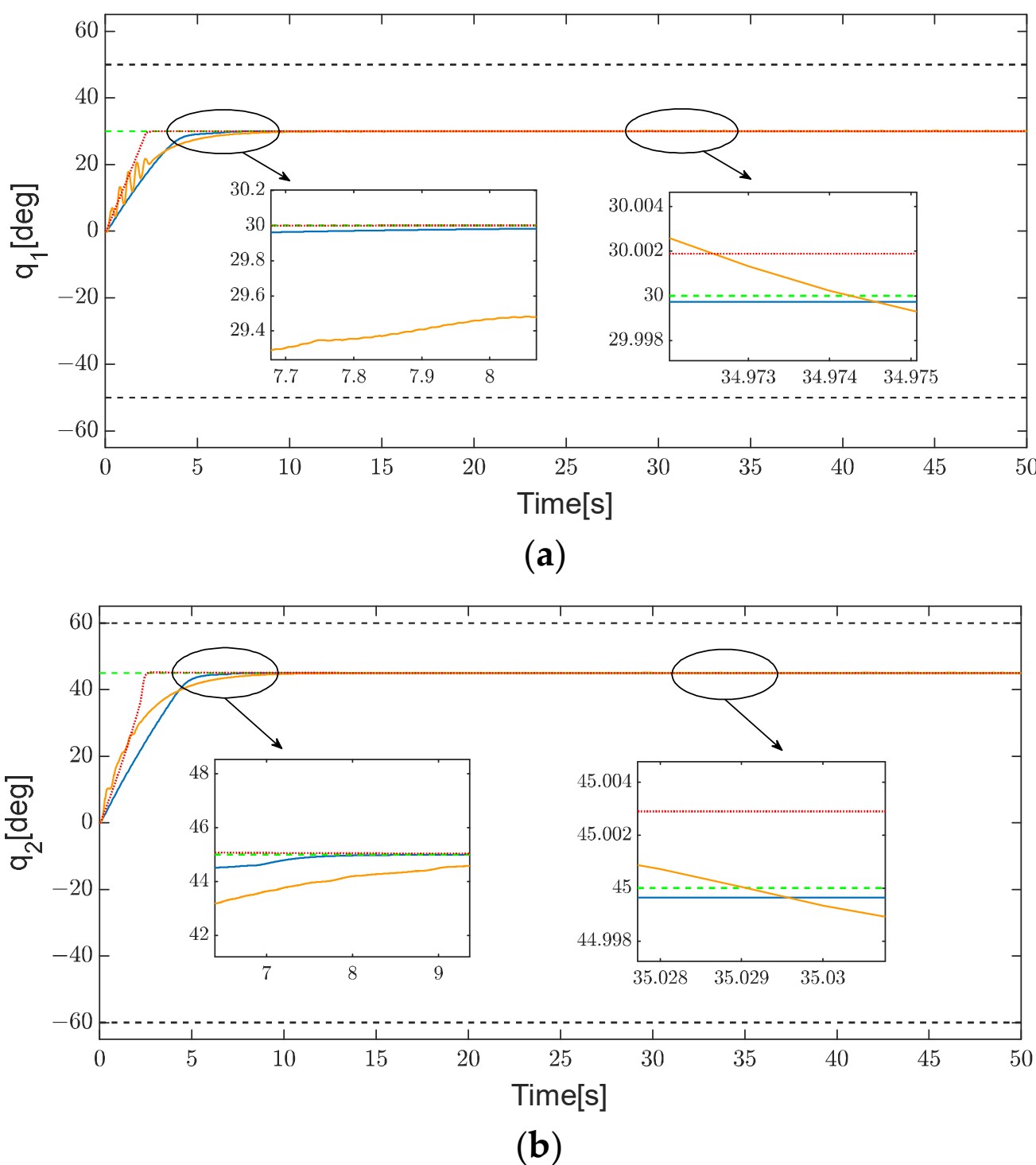

**Figure 7.** Comparison of the angular position to [43,44] in case 1: [43] scheme (yellow solid line), [44] scheme (red dash line), the proposed scheme (blue solid line), the reference angular position (green dashed line), the angular position constrains (black dashed line); (**a**) Joint 1. (**b**) Joint 2.

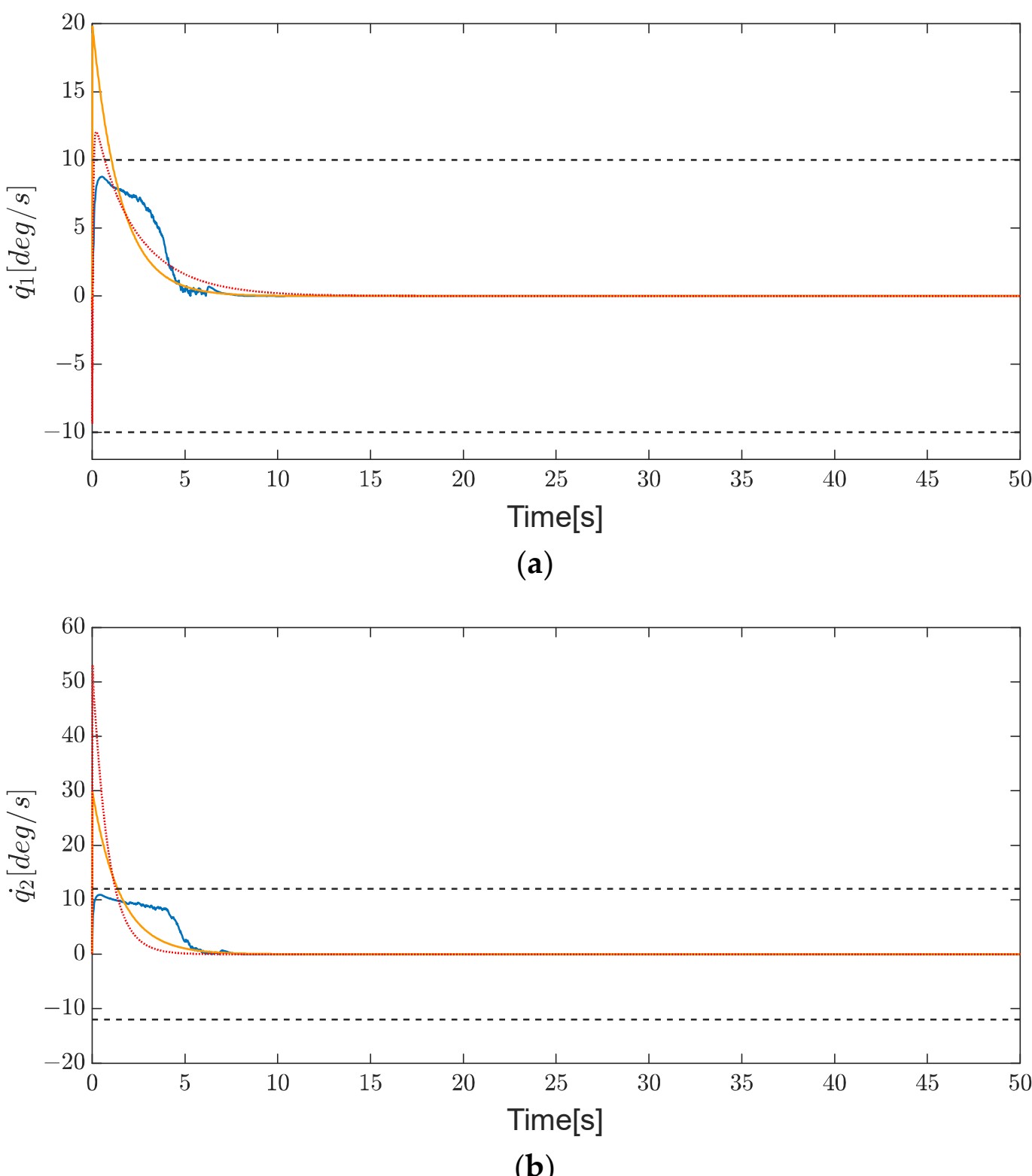

**Figure 8.** Comparison of angular velocity to [26,32] in case 1: [26] scheme (yellow solid line), [33] scheme (red dash line), the proposed scheme (blue solid line), the angular velocity constrains (black dashed line); (**a**) Joint 1. (**b**) Joint 2.

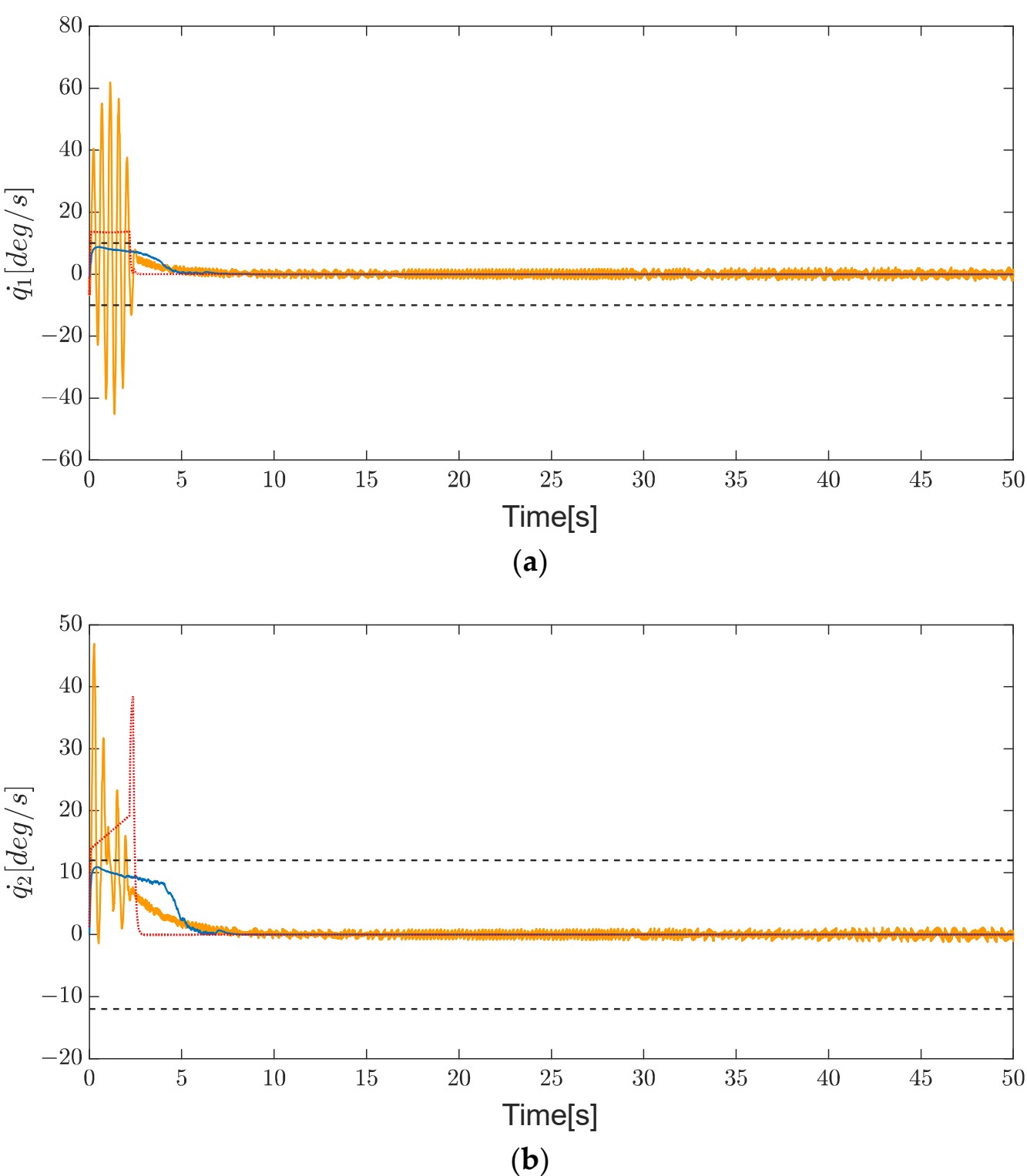

**Figure 9.** Comparison of angular velocity to [43,44] in case 1: [43] scheme (yellow solid line), [44] scheme (red dash line), the proposed scheme (blue solid line), the angular velocity constrains (black dashed line); (**a**) Joint 1. (**b**) Joint 2.

In Figures 8 and 9, the angular velocities of the proposed control scheme are within the preset constraint (red lines). While the controllers from [26,33,43,44] result in the angular velocities of two joints exceeding the constraints a $t = 0 \sim 2$ s.

**case 2: In the presence of system uncertainty, external disturbance and friction.**

In this case, the system parameters used in all the five controllers are different to that in the dynamics model of the manipulator to indicate the parametric uncertainty. Namely, $\Delta M$, $\Delta C$ and $\Delta G$ is taken as 20% of $M$, $C$ and $G$. Moreover, the friction and external disturbance, which are not known by the controllers, are applied on the dynamics model.

The friction model is from [33].

$$F(\dot{q}) = \begin{bmatrix} f_{s1}\left[\tanh\left(f_{s2}\dot{q}_1\right) - \tan\mathrm{h}\left(f_{s3}\dot{q}_1\right)\right] + f_{c1}\tanh\left(f_{c2}\dot{q}_1\right) + f_v\dot{q}_1 \\ f_{s1}\left[\tanh\left(f_{s2}\dot{q}_2\right) - \tan\mathrm{h}\left(f_{s3}\dot{q}_2\right)\right] + f_{c1}\tanh\left(f_{c2}\dot{q}_2\right) + f_v\dot{q}_2 \end{bmatrix}$$

The parameters of the friction model are given as: $f_{s1} = 20$, $f_{s2} = 5$, $f_{s3} = 3$, $f_{c1} = 10$, $f_{c2} = 2$ and $f_v = 10$.

It is widely seen to use triangular functions as the unknown disturbance in the literature of the robotic system control, such as [13,36,45,46]. Therefore, in this paper, the external disturbance is in the form of the triangular functions.

$$\tau_d = \begin{bmatrix} 0.78\sin\left(\frac{\pi}{3}t + \frac{\pi}{4}\right) + 0.065\sin\left(\frac{\pi}{10}t + \frac{\pi}{4}\right) \\ 0.58\cos\left(\frac{\pi}{3}t + \frac{\pi}{4}\right) + 0.091\sin\left(\frac{\pi}{10}t + \frac{\pi}{4}\right) \end{bmatrix}$$

The comparisons of tracking performance and tracking error in the presence of system parametric uncertainty, friction and external disturbance are shown in Figures 10–13, respectively. Clearly, the proposed controller can achieve the smallest steady state errors, which means the robustness to the lumped uncertainty and unknown disturbance. The converging rate of tracking errors of the proposed controller is faster than the controller [33,43] but slower than the controller [26,44]. It is because the preset constraint of angular velocity (black lines in Figures 14 and 15) limits the converging rate of tracking errors. Therefore, the converging rate of tracking errors could be increased by increasing the value of velocity constraint ($\Lambda_i$) (e.g., applying some better driving motors that have a greater maximum rotational speed to drive the joints of manipulator).

The computed control torques are shown in Figures 16 and 17. The chattering effect of the proposed controller occurs at the initial stage because of the two following factors: 1. the increasing value of switching gain $\hat{d}_i$ to handle disturbance at the initial stage, 2. the fuzzy Q learning mechanism tried some bad action candidates that lead to an undesirable consequence. After the initial stage ($t > 8$ s), it is observed that the proposed controller shows the smoothest control torque compared to [26,33,43,44]. Figure 18 shows the values of switching gain $\hat{d}_1$ and $\hat{d}_2$, it is clear the $\hat{d}_1$ and $\hat{d}_2$ will decrease to a small value to avoid a chattering in steady state regardless of the disturbance and uncertainty.

Notably, in both case 1 and case 2, the proposed controller can make the angular positions and angular velocities of two joints to be within their constraints during the whole period of position tracking. More precisely, the angular position of each joint is always between the two black dash lines in Figures 6, 7, 10 and 11. Meanwhile, the angular velocity of each joint is always between the two black dash lines in Figures 8, 9, 14 and 15.

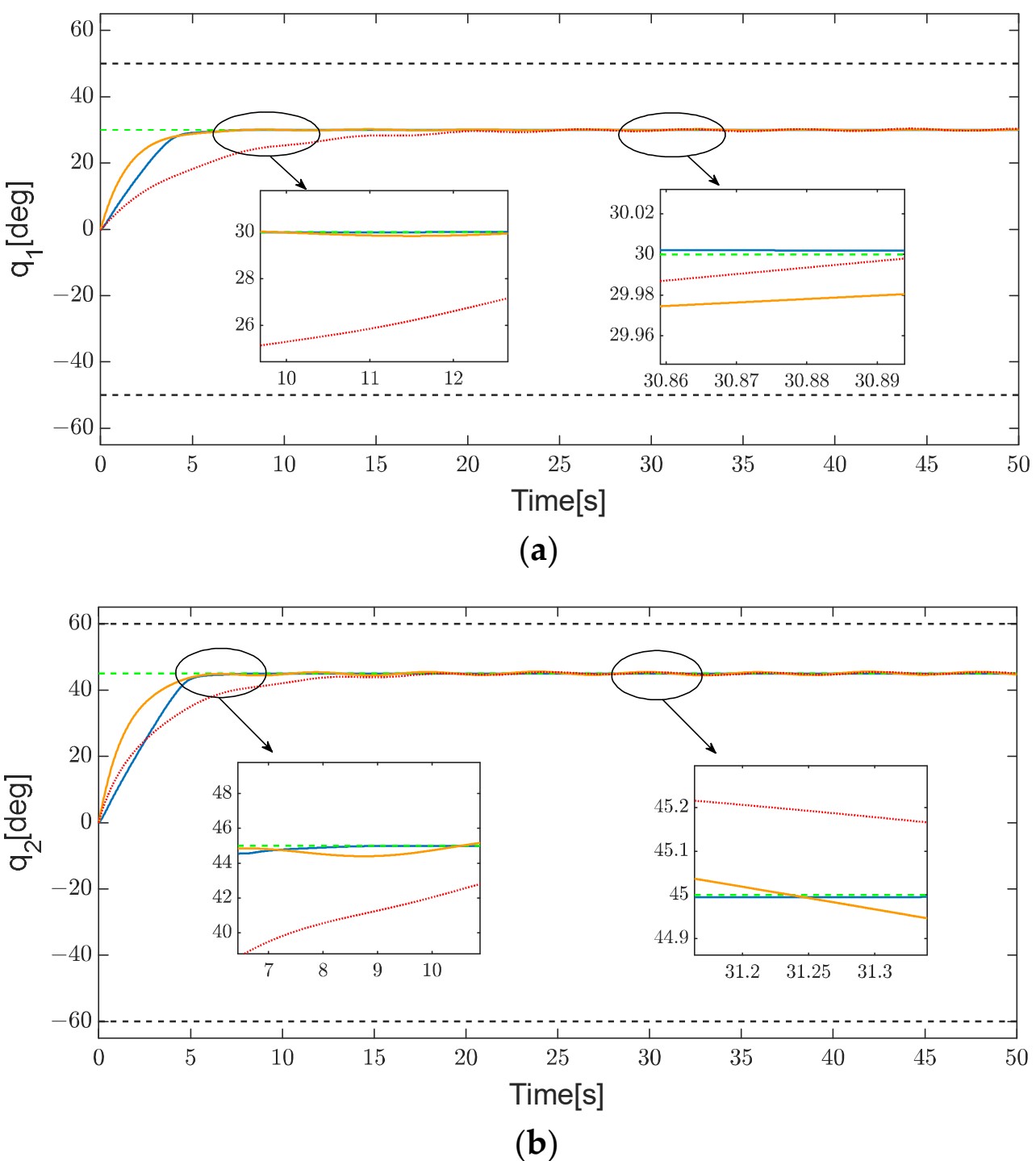

**Figure 10.** Comparison of angular position to [26,33] in case 2: the [26] scheme (yellow solid line), the [33] scheme (red dash line), the proposed scheme (blue solid line), the reference angular position (green dashed line), the angular position constrains (black dashed line); (**a**) Joint 1. (**b**) Joint 2.

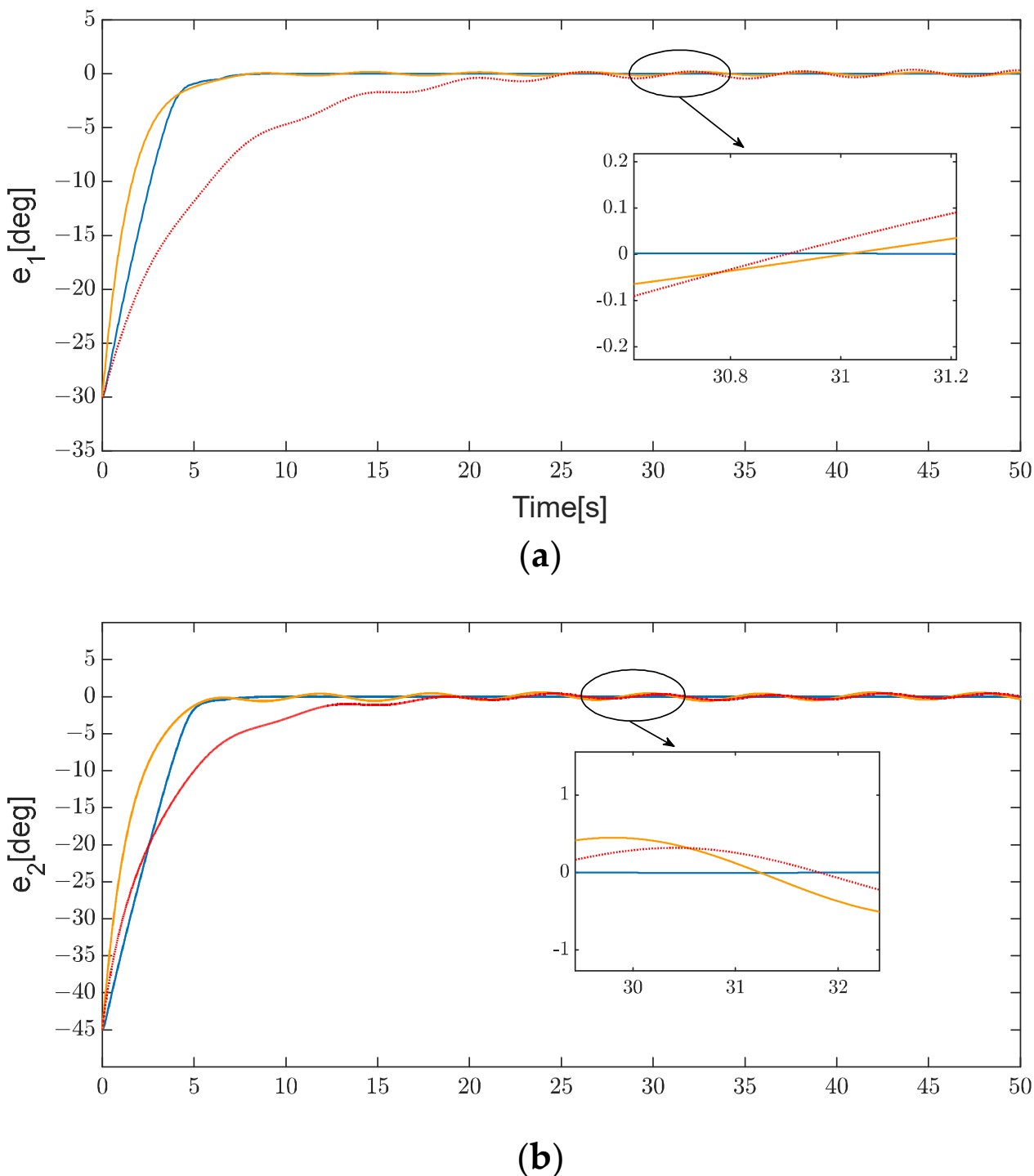

**Figure 11.** Comparison of angular position error to [26,33] in case 2: the [26] scheme (yellow solid line), the [33] scheme (red dash line), the proposed scheme (blue solid line); (**a**) Joint 1. (**b**) Joint 2.

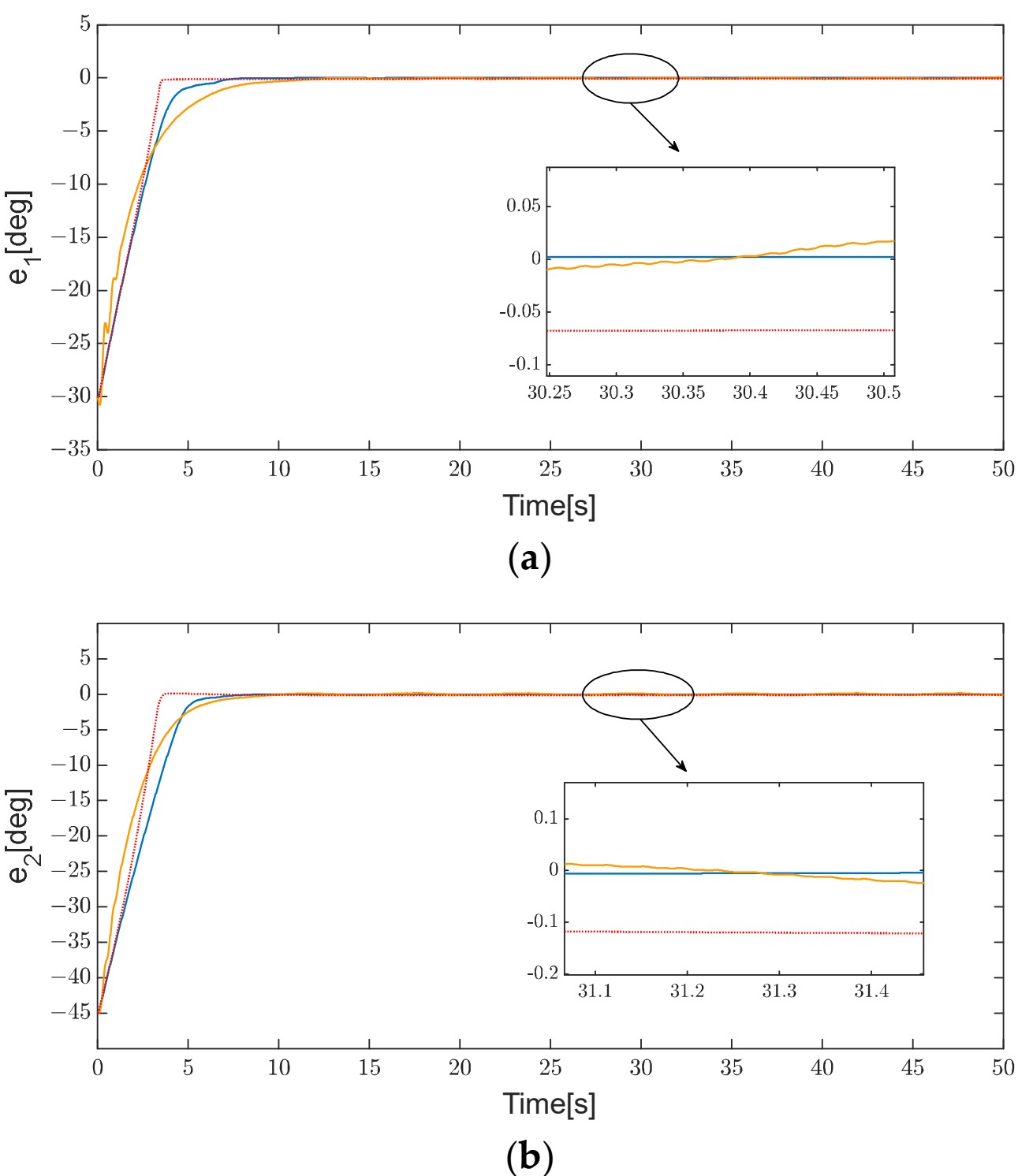

**Figure 12.** Comparison of angular position error to [43,44] in case 2: the [43] scheme (yellow solid line), the [44] scheme (red dash line), the proposed scheme (blue solid line); (**a**) Joint 1. (**b**) Joint 2.

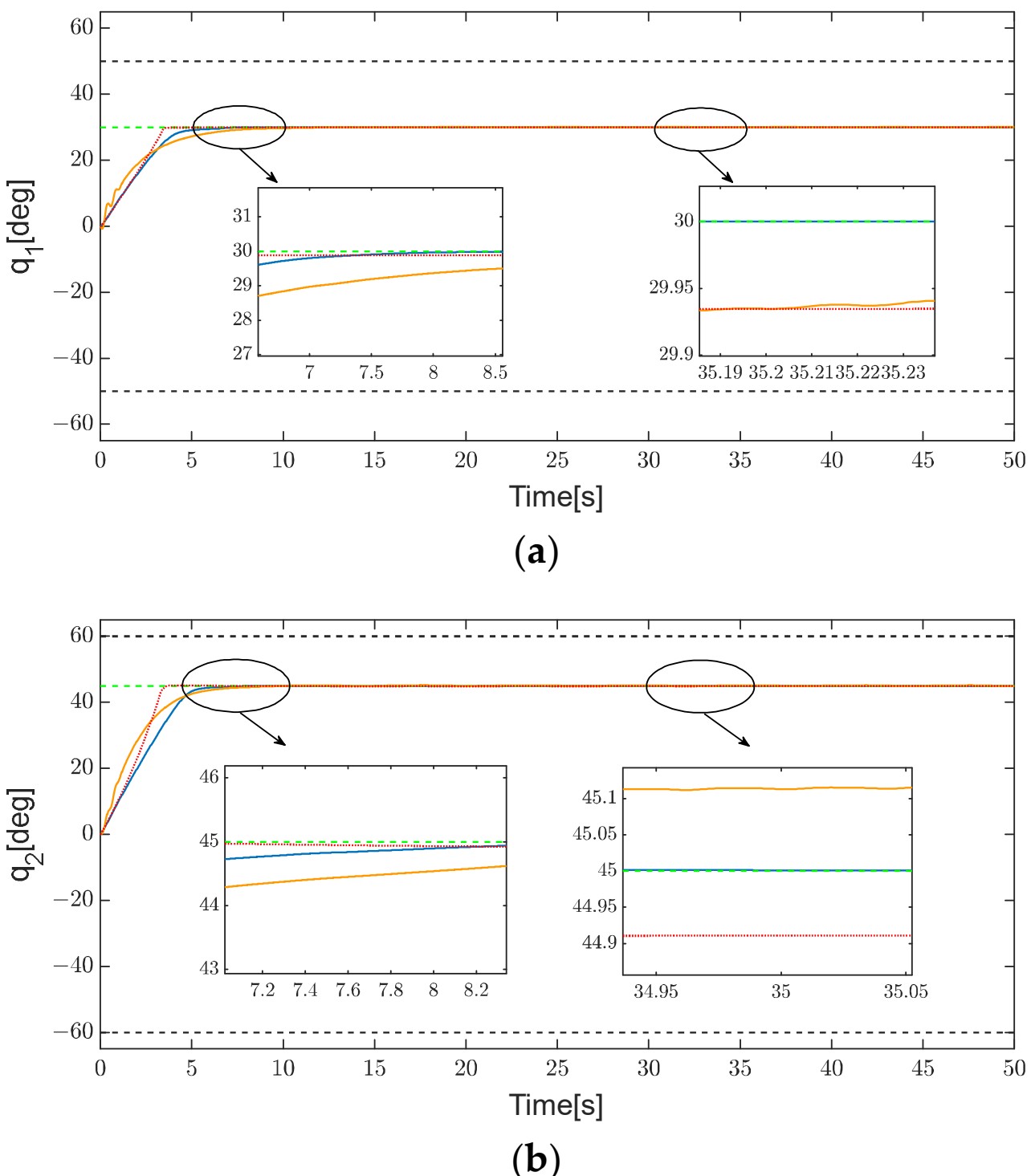

**Figure 13.** Comparison of angular position to [43,44] in case 2: The [43] scheme (yellow solid line), the [44] scheme (red dash line), the proposed scheme (blue solid line), the reference angular position (green dashed line), the angular position constrains (black dashed line); (**a**) Joint 1. (**b**) Joint 2.

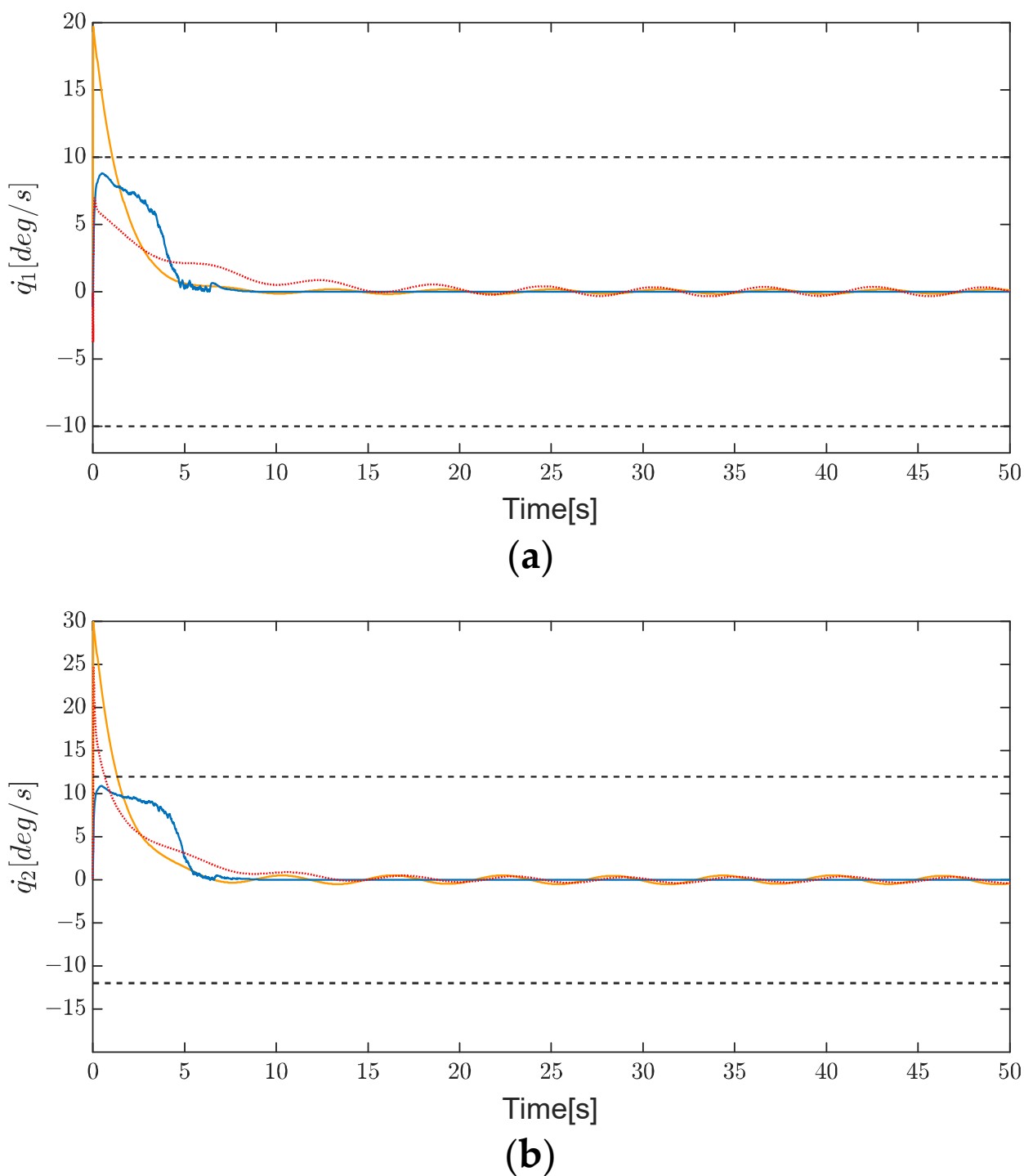

**Figure 14.** Comparison of angular velocity to [26,33] in case 2: the [26] scheme (yellow solid line), the [33] scheme (red dash line), the proposed scheme (blue solid line), the angular velocity constrains (black dashed line); (**a**) Joint 1. (**b**) Joint 2.

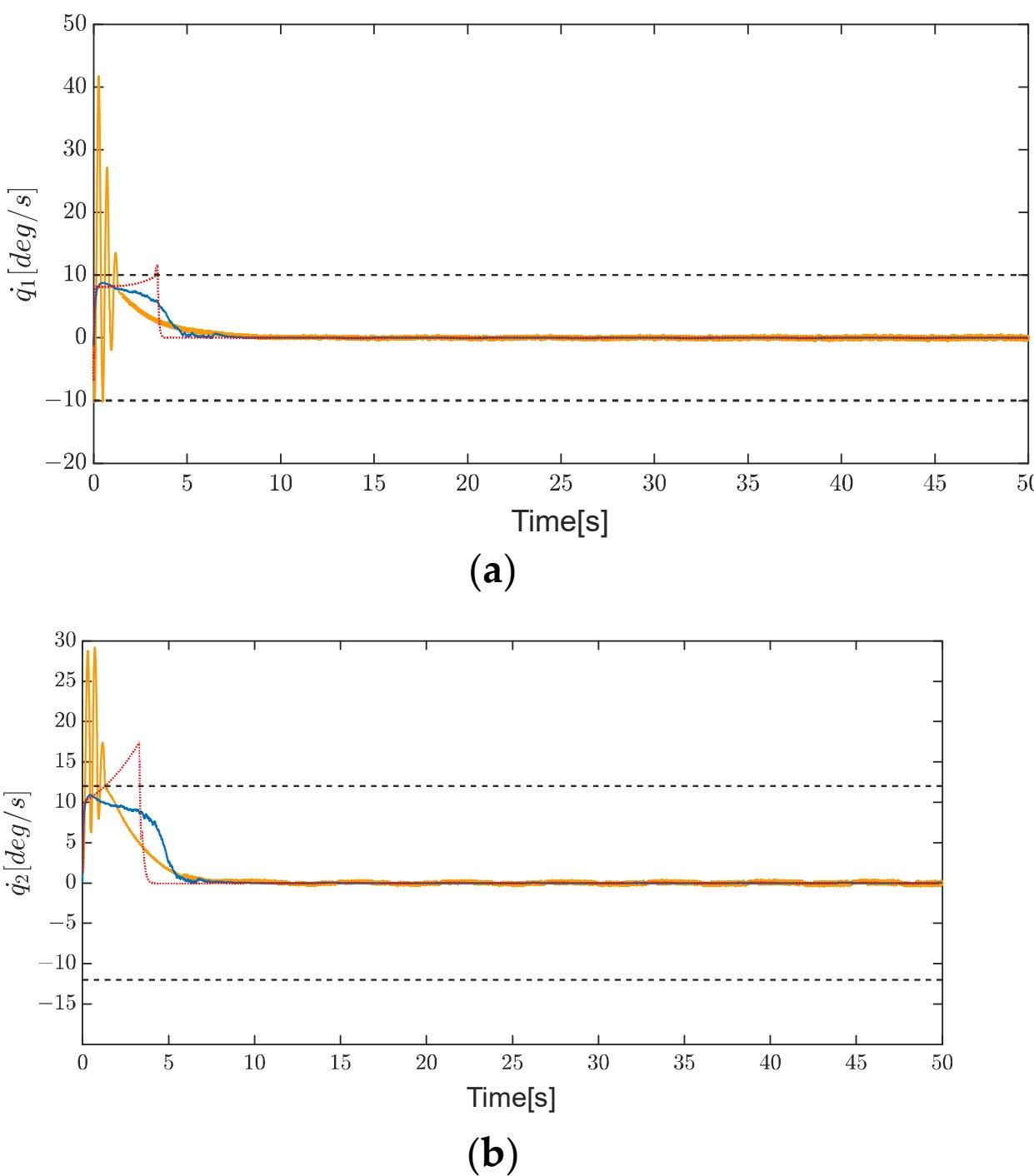

**Figure 15.** Comparison of angular velocity to [43,44] in case 2: the [43] scheme (yellow solid line), the [44] scheme (red dash line), the proposed scheme (blue solid line), the angular velocity constrains (black dashed line); (**a**) Joint 1. (**b**) Joint 2.

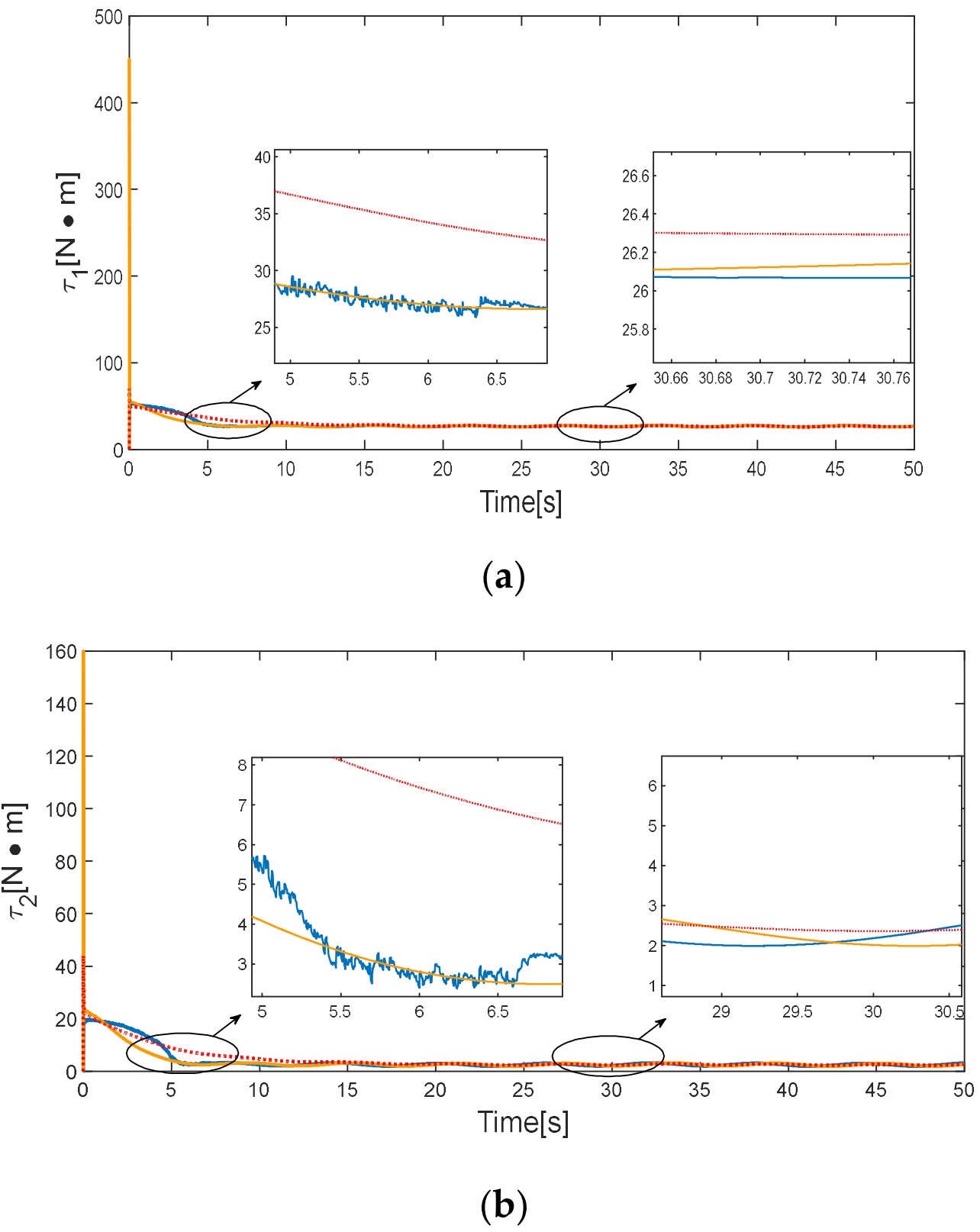

**Figure 16.** Comparison of torque to [26,33] in case 2: the [26] scheme (yellow solid line), the [33] scheme (red dash line), the proposed scheme (blue solid line); (**a**) Joint 1. (**b**) Joint 2.

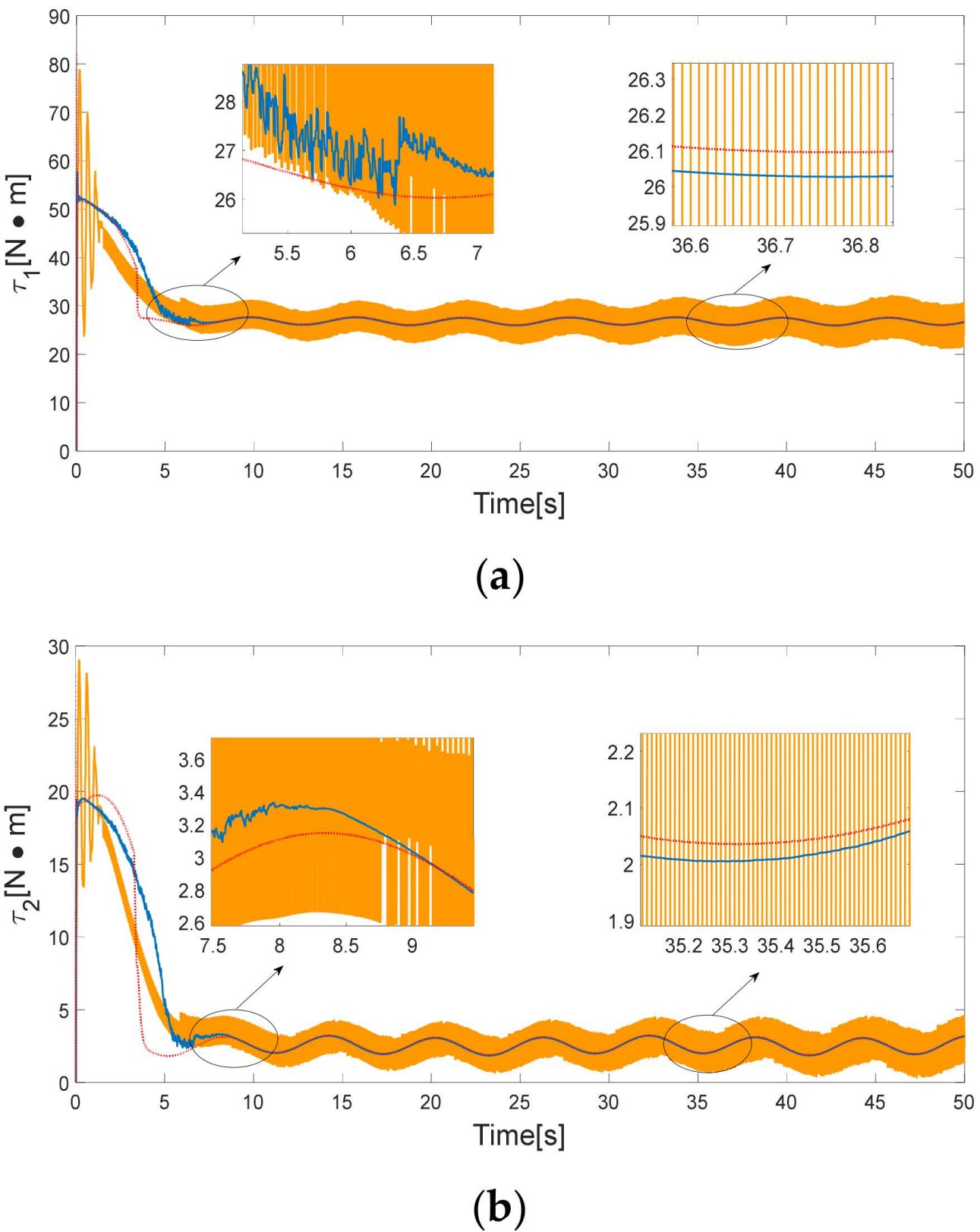

**Figure 17.** Comparison of torque to [43,44] in case 2: the [43] scheme (yellow solid line), the [44] scheme (red dash line), the proposed scheme (blue solid line); (**a**) Joint 1. (**b**) Joint 2.

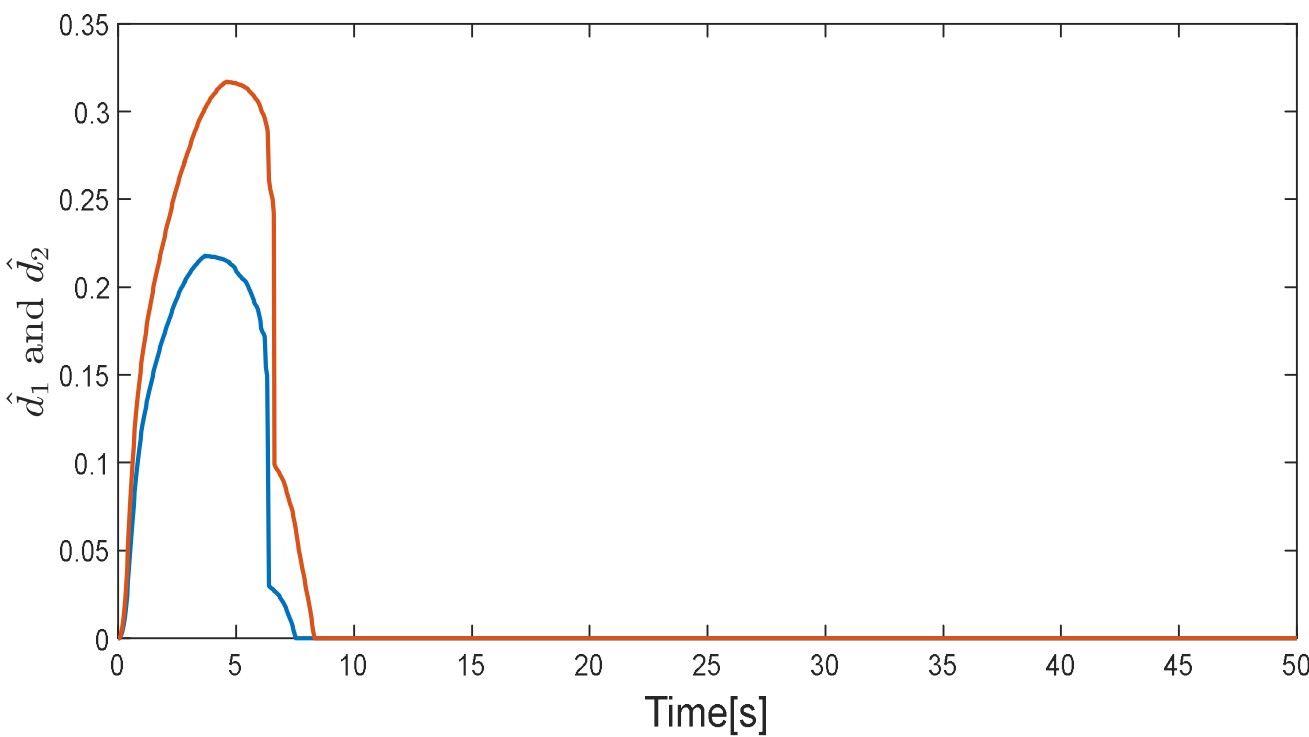

**Figure 18.** Parameter $\hat{d}_i$ handling TDE errors in case 2: Joint 1 (blue solid line). Joint 2 (orange solid line).

## 5. Conclusions

This paper proposed a novel adaptive control scheme utilizing TDE and RL for the angular position tracking control of robotic manipulators. The proposed control scheme can achieve a good tracking accuracy and a fast tracking performance even when subject to the system uncertainty and unknown disturbance. Moreover, the angular position and angular velocity of each joint of the manipulator are guaranteed to be within their preset constraints. The boundness of tracking errors and the stability of the robotic system controlled by the proposed controller are proven by Lyapunov theory. Notably, the stability will not be breached by the RL trying some bad action candidates, which ensures a safe environment for RL to explore the optimal policy. Simulation results validate the effectiveness of the proposed control scheme.

**Author Contributions:** Z.X.: Conceptualization, data curation, writing original draft. Q.L.: Formal analysis. All authors have read and agreed to the published version of the manuscript.

**Funding:** This research received no external funding.

**Institutional Review Board Statement:** Not applicable.

**Informed Consent Statement:** The author agrees to publication.

**Data Availability Statement:** The datasets generated during and/or analysed during the current study are available from the corresponding author on reasonable request.

**Conflicts of Interest:** The authors declare that there are no conflict of interest regarding the publication of this paper.

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
