# Peer review of "Reinforcement Learning-Based Adaptive Position Control Scheme for Uncertain Robotic Manipulators with Constrained Angular Position and Angular Velocity"

_applsci, doi:10.3390/app13031275_

Round 1

Reviewer 1 Report

Authors concentrate on the one approach, i.e. fuzzy one. They mention also different ones as SMC, NN and DOB. Some comparisons could beneficial. Some comparison is done only with NN, what about the others.

In the Example section, Case 1 with no uncertainty is much less imprtant than Case 2 and may be shortened to the minimum.

The results are confirmed only by simulations, which always limits the believe to them. Authors should try to do some real experiments to confirm them, which could be published later on.

Some small issues re Engleish, why do you use large letter W in Where below equations where the sentence is not finished.

Reviewer 2 Report

Please see in the attached file.

Round 2

Reviewer 1 Report

I see that now the paper is much better presented.